# Inference and reconstruction of the heimdallarchaeial ancestry of eukaryotes

Laura Eme[1,2,22], Daniel Tamarit[1,3,4,15,22], Eva F. Caceres[1,3,22], Courtney W. Stairs[1,16], Valerie De Anda[5,17], Max E. Schön[1], Kiley W. Seitz[5,18], Nina Dombrowski[5,19], William H. Lewis[1,3,20], Felix Homa[3], Jimmy H. Saw[1,21], Jonathan Lombard[1], Takuro Nunoura[6], Wen-Jun Li[7], Zheng-Shuang Hua[8], Lin-Xing Chen[9], Jillian F. Banfield[9,10], Emily St John[11], Anna-Louise Reysenbach[11], Matthew B. Stott[12], Andreas Schramm[13], Kasper U. Kjeldsen[13], Andreas P. Teske[14], Brett J. Baker[5,17] & Thijs J. G. Ettema[1,3 ✉]

In the ongoing debates about eukaryogenesis—the series of evolutionary events leading to the emergence of the eukaryotic cell from prokaryotic ancestors—members of the Asgard archaea play a key part as the closest archaeal relatives of eukaryotes[1]. However, the nature and phylogenetic identity of the last common ancestor of Asgard archaea and eukaryotes remain unresolved[2–4]. Here we analyse distinct phylogenetic marker datasets of an expanded genomic sampling of Asgard archaea and evaluate competing evolutionary scenarios using state-of-the-art phylogenomic approaches. We find that eukaryotes are placed, with high confidence, as a well-nested clade within Asgard archaea and as a sister lineage to Hodarchaeales, a newly proposed order within Heimdallarchaeia. Using sophisticated gene tree and species tree reconciliation approaches, we show that analogous to the evolution of eukaryotic genomes, genome evolution in Asgard archaea involved significantly more gene duplication and fewer gene loss events compared with other archaea. Finally, we infer that the last common ancestor of Asgard archaea was probably a thermophilic chemolithotroph and that the lineage from which eukaryotes evolved adapted to mesophilic conditions and acquired the genetic potential to support a heterotrophic lifestyle. Our work provides key insights into the prokaryote-to-eukaryote transition and a platform for better understanding the emergence of cellular complexity in eukaryotic cells.

Understanding how complex eukaryotic cells emerged from prokaryotic ancestors represents a major challenge in biology[1,5]. A main point of contention in refining eukaryogenesis scenarios revolves around the exact phylogenetic relationship between Archaea and eukaryotes. The use of phylogenomic approaches with improved models of sequence evolution combined with enhanced archaeal taxon sampling—progressively uncovered using metagenomics—has recently produced strong support for the two-domain tree of life, in which the eukaryotic clade branches from within Archaea[6–10]. The discovery of the first Lokiarchaeia genome provided additional evidence for the two-domain topology because this lineage was shown to represent, at the time, the closest relative of eukaryotes in phylogenomic analyses[2]. Moreover, Lokiarchaeia genomes specifically contain many genes that encode eukaryotic signature proteins (ESPs)—proteins involved in hallmark complex processes of the eukaryotic cell—more so than any other prokaryotic lineage. The subsequent identification and analyses of several diverse relatives of Lokiarchaeia, together forming the Asgard archaea superphylum, confirmed that Asgard archaea represent the closest archaeal relatives of eukaryotes[1–3]. Their exact evolutionary relationship to eukaryotes, however, remained unresolved. Specially, it has

[1]Department of Cell and Molecular Biology, Science for Life Laboratory, Uppsala University, Uppsala, Sweden. [2]Laboratoire Écologie, Systématique, Évolution, CNRS, Université Paris-Saclay, AgroParisTech, Gif-sur-Yvette, France. [3]Laboratory of Microbiology, Wageningen University and Research, Wageningen, The Netherlands. [4]Department of Aquatic Sciences and Assessment, Swedish University of Agricultural Sciences, Uppsala, Sweden. [5]Department of Marine Science, Marine Science Institute, University of Texas Austin, Port Aransas, TX, USA. [6]Research Center for Bioscience and Nanoscience (CeBN), Japan Agency for Marine-Earth Science and Technology (JAMSTEC), Yokosuka, Japan. [7]State Key Laboratory of Biocontrol, Guangdong Provincial Key Laboratory of Plant Resources and Southern Marine Science and Engineering Guangdong Laboratory (Zhuhai), School of Life Sciences, Sun Yat-Sen University, Guangzhou, PR China. [8]Chinese Academy of Sciences Key Laboratory of Urban Pollutant Conversion, Department of Environmental Science and Engineering, University of Science and Technology of China, Hefei, PR China. [9]Department of Earth and Planetary Sciences, University of California, Berkeley, CA, USA. [10]Department of Environmental Science, Policy, and Management, University of California, Berkeley, CA, USA. [11]Department of Biology, Portland State University, Portland, OR, USA. [12]School of Biological Sciences, University of Canterbury, Christchurch, New Zealand. [13]Section for Microbiology, Department of Biology, Aarhus University, Aarhus, Denmark. [14]Department of Earth, Marine and Environmental Sciences, University of North Carolina, Chapel Hill, NC, USA. [15]Present address: Theoretical Biology and Bioinformatics, Department of Biology, Faculty of Science, Utrecht University, Utrecht, The Netherlands. [16]Present address: Department of Biology, Lund University, Lund, Sweden. [17]Present address: Department of Integrative Biology, University of Texas Austin, Austin, TX, USA. [18]Present address: Structural and Computational Biology, European Molecular Biology Laboratory, Heidelberg, Germany. [19]Present address: Department of Marine Microbiology and Biogeochemistry, NIOZ, Royal Netherlands Institute for Sea Research, AB Den Burg, The Netherlands. [20]Present address: Department of Biochemistry, University of Cambridge, Cambridge, UK. [21]Present address: Department of Biological Sciences, The George Washington University, Washington, DC, USA. [22]These authors contributed equally: Laura Eme, Daniel Tamarit, Eva F. Caceres. ✉e-mail: thijs.ettema@wur.nl

been unclear whether eukaryotes evolved from within Asgard archaea or whether they represented a sister lineage[3]. Furthermore, two studies questioned this view of the tree of life altogether, suggesting that Asgard archaea represent a deep-branching Euryarchaea-related clade[11,12]. These studies suggested that, in accordance with the three-domain tree, eukaryotes represent a sister group to all Archaea; however, this view has been challenged[13,14]. More recently, a study that included an expanded taxonomic sampling of Asgard archaeal genome data failed to resolve the phylogenetic position of eukaryotes in the tree of life[4].

Here we expand the genomic diversity of Asgard archaea by generating 63 new Asgard archaeal metagenome-assembled genomes (MAGs) from samples obtained from 11 locations around the world. By analysing the enlarged genomic sampling of Asgard archaea using state-of-the-art phylogenomics analyses, including recently developed gene tree and species tree reconciliation approaches for ancestral genome content reconstruction, we firmly place eukaryotes as a clade nested within the Asgard archaea. By revealing key features regarding the identity, nature and physiology of the last Asgard archaea and eukaryotes common ancestor (LAECA), our results represent important, thus far missing pieces of the eukaryogenesis puzzle.

## Expanded Asgard archaea genome diversity

To increase the genomic diversity of Asgard archaea, we sampled aquatic sediments and hydrothermal deposits from 11 geographically distinct sites (Supplementary Table 1 and Supplementary Fig. 1). After extraction and sequencing of total environmental DNA, we assembled and binned metagenomic contigs into MAGs. Of these MAGs, 63 belonged to the Asgard archaea superphylum, with estimated median completeness and redundancy values of 83% and 4.2%, respectively (Supplementary Table 1). To assess the genomic diversity in this dataset, we reconstructed a phylogeny of ribosomal proteins encoded in a conserved 15 ribosomal protein (RP15) gene cluster from these MAGs and in all publicly available Asgard archaea assemblies (retrieved 29 June 2021; Fig. 1). These analyses showed that we expanded the genomic sampling across previously described major Asgard archaea clades (that is, Lokiarchaeia, Thorarchaeia, Heimdallarchaeia, Odinarchaeia, Hermodarchaeia, Sifarchaeia, Jordarchaeia and Baldrarchaeia[2–4,15,16]). We also recovered a previously undescribed clade of high taxonomic rank (*Candidatus* Asgardarchaeia; see Extended Data Fig. 1 and Supplementary Information for a proposed uniformization of Asgard archaea taxonomic classification to which we will adhere throughout the current paper). We observed that the median estimated Asgard archaeal genome size (3.8 Mb) is considerably larger than those of representative genomes from TACK archaea and Euryarchaea (median = 1.8 Mb for both) and DPANN archaea (median = 1.2 Mb) (Supplementary Table 1). Among Asgard archaea, Odinarchaeia displayed the smallest genomes (median = 1.4 Mb), whereas Lokiarchaeales and Helarchaeales contained the largest (median = 4.3 Mb for both). Unlike other major Asgard archaeal clades, Heimdallarchaeia possessed a wide range of genome sizes, spanning from 1.6 to 7.4 Mb (median = 3.5 Mb). This large class contained five clades with diverse features: Njordarchaeales (median genome size = 2.4 Mb); Kariarchaeaceae (median genome size = 2.7 Mb); Gerdarchaeales (median genome size = 3.4 Mb); Heimdallarchaeaceae (median genome size = 3.7 Mb); and Hodarchaeales (median genome size = 5.1 Mb). The smallest heimdallarchaeial genome corresponded to the only Asgard archaeal MAG recovered from a marine surface water metagenome (Heimdallarchaeota archaeon RS678)[17]. This result is in agreement with the reduced genome sizes typically observed among prokaryotic plankton of the euphotic zone[18].

## Identification of phylogenetic conflict

Inferring deep evolutionary relationships in the tree of life is considered one of the hardest problems in phylogenetics. To interrogate the evolutionary relationships within the current set of Asgard archaeal phyla, and between Asgard archaea and eukaryotes, we performed an exhaustive range of phylogenomic analyses. We analysed a pre-existing marker dataset comprising 56 concatenated ribosomal protein sequences (RP56)[2,3] for a phylogenetically diverse set of 331 archaeal (175 Asgard archaea, 41 DPANN archaea, 43 Euryarchaea and 72 TACK archaea representatives) and 14 eukaryotic taxa (Supplementary Table 2). Of note, the inclusion of an expanded diversity of 12 new Korarchaeota MAGs among these TACK archaea considerably affected phylogenomic analyses (see below). Initial maximum-likelihood (ML) phylogenetic inference based on this RP56 dataset confirmed the existence of 12 major Asgard archaeal clades of high taxonomic rank (Supplementary Fig. 2). These included the previously described Lokiarchaeia, Odinarchaeia, Heimdallarchaeia and Thorarchaeia[2,3], for which we present 36 new genomes here. The clades also included the recently proposed Sifarchaeia[16], Hermodarchaeia[15], Jordarchaeia[19], Wukongarchaeia[4] and Baldrarchaeia[4], for most of which we also identified new near-complete MAGs. Finally, we identified 15 MAGs that represented the recently described Njordarchaeales[20] (which we show below is a divergent candidate order within Heimdallarchaeia, see below) and a single MAG that represented a new candidate class, Asgardarchaeia (described elsewhere) (Fig. 1). Notably, careful inspection of the obtained RP56 tree uncovered a potential artefact: Njordarchaeales, considered bona fide Asgard archaea based on the presence of many encoded typical Asgard archaeal ESPs[3], branched outside Asgard archaea, at the base of the TACK superphylum and as a sister lineage to Korarchaeota in the RP56 tree. In addition, eukaryotes branched at the base of the clade formed by Korarchaeota and Njordarchaeales, albeit with weak support. Hereafter, we focused on disentangling the historically correct phylogenetic signal from noise and potential artefacts.

## Alternative phylogenomic markers

Despite often being used in phylogenomic analyses, ribosomal proteins have been suggested to contribute to phylogenetic artefacts owing to inherent compositional sequence biases[21,22]. Our results revealed a placement of eukaryotes inconsistent with previous analyses, the previously mentioned incoherent placement of Njordarchaeales and the presence of long branches at the base of both of these clades in the RP56 tree. Therefore, we sought to use an alternative phylogenetic marker set to obtain a stable Asgard archaeal species tree and to further investigate the phylogenetic position of eukaryotes. We constructed an independent new marker dataset comprising 54 proteins of archaeal origin in eukaryotes (NM54 dataset; Methods). The NM54 proteins are mostly involved in diverse informational, metabolic and cellular processes, but do not include ribosomal proteins (Supplementary Table 2). These proteins are longer and therefore putatively more phylogenetically informative compared with the RP56 markers. Moreover, the broader functional distribution of NM54 markers is less likely to cause phylogenetic reconstruction artefacts induced by strong co-evolution between proteins—something that is to be expected for functionally and structurally cohesive ribosomal proteins[23]. If co-evolving protein sequences are compositionally biased, then they would violate evolutionary model assumptions of fixed composition across species. Consequently, their concatenation is expected to strengthen the artefactual, non-phylogenetic signal and the statistical support for incorrect relationships[24]. We therefore decided to independently evaluate the concatenated NM54 and RP56 marker datasets for downstream phylogenomic analyses. We observed that ML phylogenomic analyses of the NM54 dataset recovered Njordarchaeales as bona fide Asgard archaea and placed them as the closest relatives of eukaryotes (bootstrap support, BS = 99%; Supplementary Fig. 3), as was proposed in a recent analysis[20]. We set out to investigate the underlying causes for the contradictory results between the NM54 and RP56 datasets. To that end, we first assessed the effect of taxon sampling on phylogenetic reconstructions by removing eukaryotic

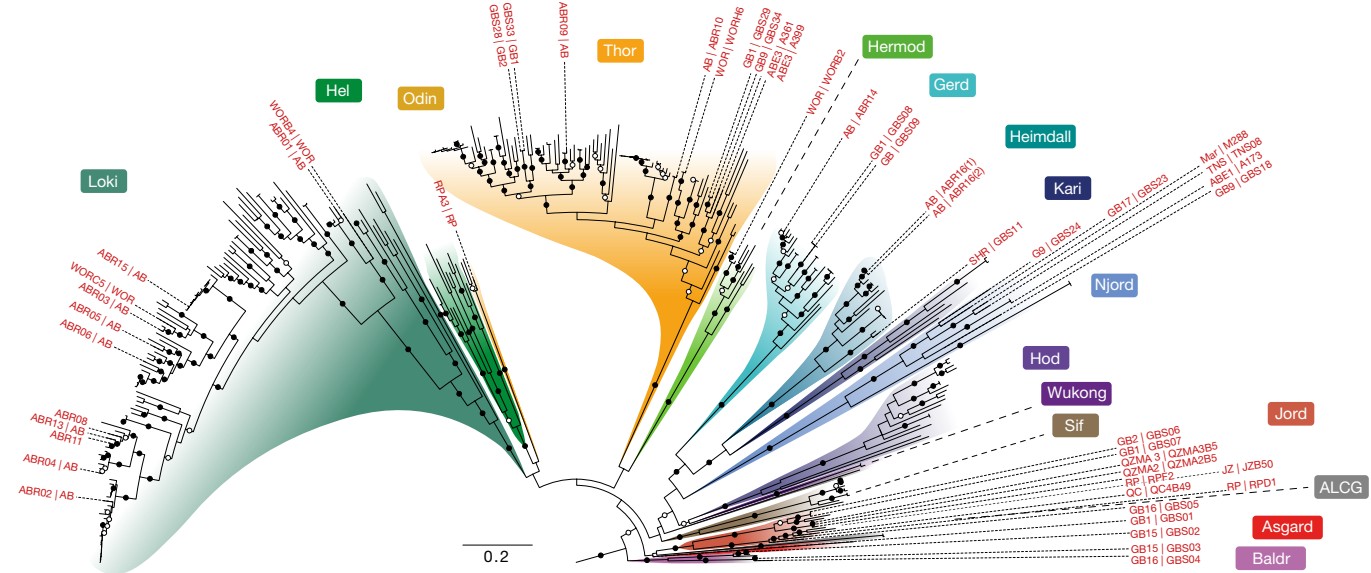

**Fig. 1 | Phylogenomic analysis of 15 concatenated ribosomal proteins expands Asgard archaea diversity.** ML tree (IQ-TREE, WAG+C60+R4+F+PMSF model) of concatenated protein sequences from at least 5 genes, encoded on a single contig, of a RP15 gene cluster retrieved from publicly available and newly reported Asgard archaeal MAGs. Bootstrap support (100 pseudo-replicates) is indicated by circles at branches, with filled and open circles representing values equal to or larger than 90% and 70% support, respectively. Leaf names indicate the geographical source and isolate name (inner and outer label, respectively) for the MAGs reported in this study. Only the in-group is shown (263 out of 542 total sequences). Scale bar denotes the average number of substitutions per site. AB, Aarhus Bay (Denmark); ABE, ABE vent field, Eastern Lau Spreading Center; ALCG, Asgard Lake Cootharaba Group; Asgard, Asgardarchaeia; Baldr, Baldrarchaeia; GB, Guaymas Basin (Mexico); Gerd, Gerdarchaeales; Hel, Helarchaeales; Heimdall, Heimdallarchaeaceae; Hermod, Hermodarchaeia; Hod, Hodarchaeales; Jord, Jordarchaeia; JZ, Jinze (China); Kari, Kariarchaeaceae; Loki, Lokiarchaeales; Mar, Mariner vent field, Eastern Lau Spreading Center; Njord, Njordarchaeales; Odin, Odinarchaeia; QC, QuCai village (China); QZM, QuZhuoMu village (China); RP, Radiata Pool (New Zealand); SHR, South Hydrate Ridge; Sif, Sifarchaeia; Thor, Thorarchaeia; TNS, Taketomi Island (Japan); WOR: White Oak River (USA); Wukong, Wukongarchaeia.

and/or DPANN and/or Korarchaeota sequences from the alignments. This was done for two main reasons: (1) eukaryotes and DPANN archaea represent long-branching clades that potentially induce long-branch attraction artefacts; and (2) we wanted to investigate the effects of removing eukaryotes and Korarchaeota, which were the sister lineages of Njordarchaeales in the NM54 and RP56 phylogenetic analyses, respectively. Following this, we recoded the alignments into four states (using SR4 recoding[25]) to ameliorate potential phylogenetic artefacts arising from model misspecification at mutationally saturated or compositionally biased sites[14,26–28]. Furthermore, with a similar goal, we applied a fast-evolving site removal (FSR) procedure to the concatenated datasets, as fast-evolving sites are often mutationally saturated. We performed phylogenetic analyses of the abovementioned datasets in both ML and Bayesian inference (BI) frameworks under sophisticated evolutionary models that account for sequence heterogeneity in the substitution process across sites (mixture models; Supplementary Table 2).

Phylogenomic analyses of the abovementioned combinations of taxon sampling, data treatments and phylogenetic frameworks revealed that Njordarchaeales are artefactually attracted to Korarchaeota in RP56 datasets (Supplementary Information). This attraction is likely to be caused by the high compositional similarity of njordarchaeal and korarchaeal RP56 ribosomal protein sequences, which is probably linked to their shared hyperthermophilic lifestyle (Supplementary Figs. 4–6). Analyses of RP56 datasets from which Korarchaeota were removed recovered Njordarchaeales as an order at the base of or within Heimdallarchaeia (Supplementary Fig. 7). This result was consistent with phylogenomic analyses of the NM54 dataset that included Korarchaeota (Supplementary Fig. 3). Next, in our efforts to resolve the phylogenetic placement of eukaryotes, we initially performed phylogenomic analyses on variations of the RP56 and NM54 datasets (Supplementary Table 2 and Discussion). However, compared with the RP56 dataset, the NM54 dataset is larger and less compositionally biased and is therefore expected to have retained a stronger historical phylogenetic signal. Consequently, we focused the rest of our discussion on this more reliable dataset.

## Eukarya emerged within Heimdallarchaeia

Subsequent phylogenetic analyses of untreated NM54 datasets with diverse taxon sampling variations recovered eukaryotes as a sister clade to Njordarchaeales in ML analyses (Supplementary Fig. 3, Supplementary Table 2 and Supplementary Information). However, ML analyses of the SR4-recoded datasets showed very weak statistical support for this position, strongly suggesting that the previously observed phylogenetic affiliation between Njordarchaeales and eukaryotes could represent an artefact. Furthermore, when both SR4-recoding and FSR treatments were combined, eukaryotes were nested within Heimdallarchaeia as a sister group to the order Hodarchaeales, with as little as 10% of fast-evolving sites removed (Fig. 2 and Supplementary Fig. 8). This position was supported by ML analyses of NM54 datasets across all taxon selection variations (removing DPANN archaea and/or Korarchaeota and/or Njordarchaeales). Congruently, the monophyly of eukaryotes and Hodarchaeales was systematically recovered by BI of recoded and FSR datasets (in combination or not; Fig. 2, Supplementary Table 2). In addition, the position of Njordarchaeales shifted during these analyses, moving from a deep position at the base of Heimdallarchaeia and Wukongarchaeia to a more nested position, forming a clade with Gerdarchaeales, Kariarchaeaceae, and Heimdallarchaeaceae (Supplementary Discussion). This shift was observed in analyses of both the NM54 and the RP56 datasets when SR4 recoding and FSR was combined (Supplementary Figs. 9 and 10). This result provides support for the idea that Njordarchaeales represent a divergent order-level lineage of Heimdallarchaeia.

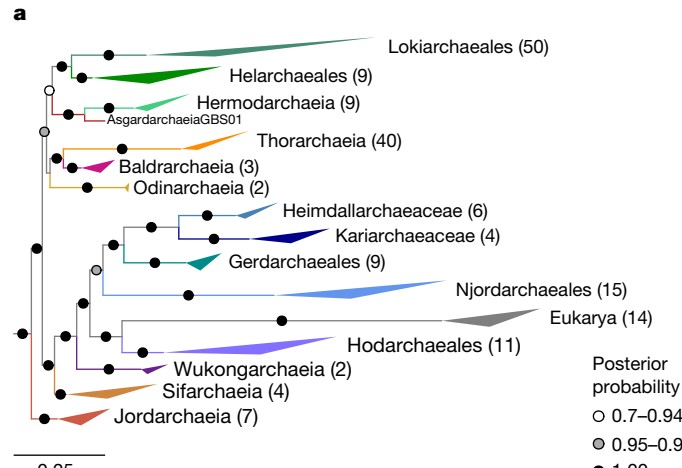

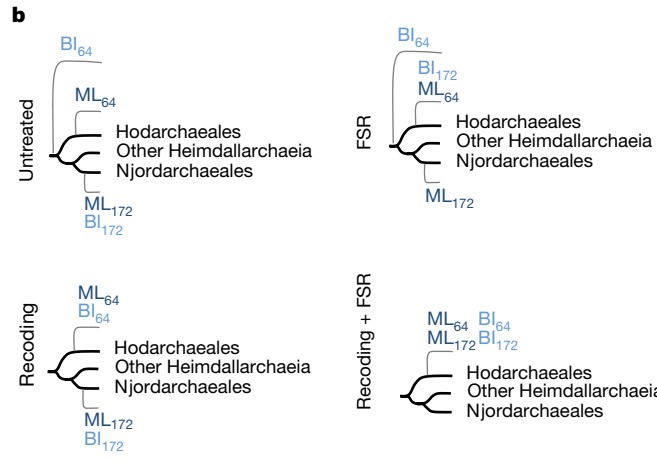

**Fig. 2 | Phylogenomic analyses based on 54 concatenated non-ribosomal proteins support the emergence of eukaryotes as sister to Hodarchaeales. a**, BI based on 313 archaeal taxa, using Euryarchaea, TACK and DPANN archaea as the outgroup (not shown) (NM54-A_sr4 alignment, 13,513 amino acid positions). The concatenation was SR4-recoded and analysed using the CAT+GTR model (4 chains, ~21,000 generations). **b**, Schematic representation of the shift in the position of eukaryotes (grey branches) in ML and BI analyses of this dataset under different treatments. Untreated: unprocessed dataset;

Recoding: SR4-recoded dataset; FSR: Fast-Site Removal; Recoding+Fast-Site Removal: Fast-site removal combined with SR4-recoding (the topology most often recovered after removing 10% to 50% fastest-evolving sites, in steps of 10%, is shown). 172 and 64 refer to phylogenomic datasets containing 172 and 64 Asgard archaea, respectively. For detailed results of phylogenomic analyses, see Supplementary Table 3. Scale bar denotes the average expected number of substitutions per site.

In summary, resolving the position of eukaryotes relative to Asgard archaea is not trivial (Supplementary Discussion). In our efforts to extract the historically correct phylogenetic signal, we provide support for eukaryotes forming a well-nested clade within the Asgard archaea phylum, consistent with the two-domain tree of life scenario. Specifically, we observed that eukaryotes affiliate with the Heimdallarchaeia in analyses in which we systematically reduced phylogenetic artefacts, predominantly converging on a position of eukaryotes as sister to Hodarchaeales. This finding is also in line with the observed ESP content and genome evolution dynamics (see below).

## Informational ESPs in Hodarchaeales

Most of the ESPs previously identified in a limited sampling of Asgard archaea[2,3] are widespread across all the Asgard archaeal classes included in the current study (Fig. 3 and Supplementary Table 3). Notably, we observed the following exceptions in support of the phylogenetic affiliation between Hodarchaeales and eukaryotes, particularly among ESPs involved in information processing. (1) the ε DNA polymerase subunit is only found in Hodarchaeales. (2) Ribosomal protein L28e (including Mak16) homologues are specific to Njordarchaeales and Hodarchaeales members. (3) Many archaea that lack genes encoding proteins for the synthesis of diphthamide, a modified histidine residue that is specifically present in archaeal and eukaryotic elongation factor 2 (EF-2), instead encode a second EF-2 paralogue that misses key residues required for diphthamide modification[29]. Notably, we found that among all Asgard archaea, only MAGs of all sampled Hodarchaeales members have *dph* genes in addition to a single gene encoding canonical EF-2, which branches at the base of their eukaryotic counterparts in phylogenetic analyses (Supplementary Fig. 11 and Supplementary Information). (4) Although RPL22e and RNA polymerase subunit RPB8 are found in several Asgard archaeal phyla, the only Heimdallarchaeia genomes that have these genes are members of the Hodarchaeales. Finally, (5) we identified amino-terminal histone tails characteristic of eukaryotic histones in all three Hodarchaeales MAGs and in three Njordarchaeales genomes (Supplementary Information). Altogether, the identification of these key informational ESPs, in agreement with results from the phylogenomic analyses described above, supports

the idea that Hodarchaeales represent the closest archaeal relatives of eukaryotes.

## Expanded set of translocon-linked ESPs

In our search for putative new ESPs in the expanded Asgard archaeal genomic diversity, we uncovered several additional homologues of proteins associated with the eukaryotic translocon. This protein complex is primarily responsible for the post-translational modification of proteins and subsequent insertion into or transport across the membrane of the endoplasmic reticulum (ER)[30]. The eukaryotic translocon is composed of the core Sec61 protein-conducting channel and several accessory components. These include the oligosaccharyltransferase (OST) and translocon-associated protein (TRAP) complexes (Extended Data Fig. 2), both of which are involved in the biogenesis of N-glycosylated proteins[31]. The TRAP complex is composed of two to four subunits in eukaryotes. Using distant-homology detection methods, we identified homologues of three of these subunits that were broadly distributed across Asgard archaeal genomes, whereas the fourth one was detected only in a few thorarchaeial MAGs (Fig. 3). The eukaryotic OST complex generally comprises six to eight subunits organized into three subcomplexes that are collectively embedded in the ER membrane[32] (Extended Data Fig. 2). Apart from STT3 (also known as AglB) (OST subcomplex-II), which represents the catalytic subunit and is universally found across all three domains of life, other OST subcomplexes generally do not possess prokaryotic homologues beyond the Ost1 (also known as ribophorin I) (OST subcomplex-I) and Ost3 (also known as Tusc3) (OST subcomplex-II) subunits previously reported in Asgard archaea[3]. Here we report the identification of Asgard archaeal homologues of all five additional subunits: Ost2 (also known as Dad1); Ost4; Ost5 (also known as TMEM258); SWP1 (also known as ribophorin II); and WBP1 (also known as Ost48). We identified homologues of Ost4 and Ost5 (OST subcomplex-I) in most Asgard archaeal classes. Ost2, WBP1 and Swp1, to our knowledge, are the first subcomplex-III subunits described in prokaryotes. The distribution of these subunits was restricted to Heimdallarchaeia, including Njordarchaeales for WBP1, thereby further supporting their monophyly. Our findings indicate that Asgard archaea and, by inference, LAECA, potentially

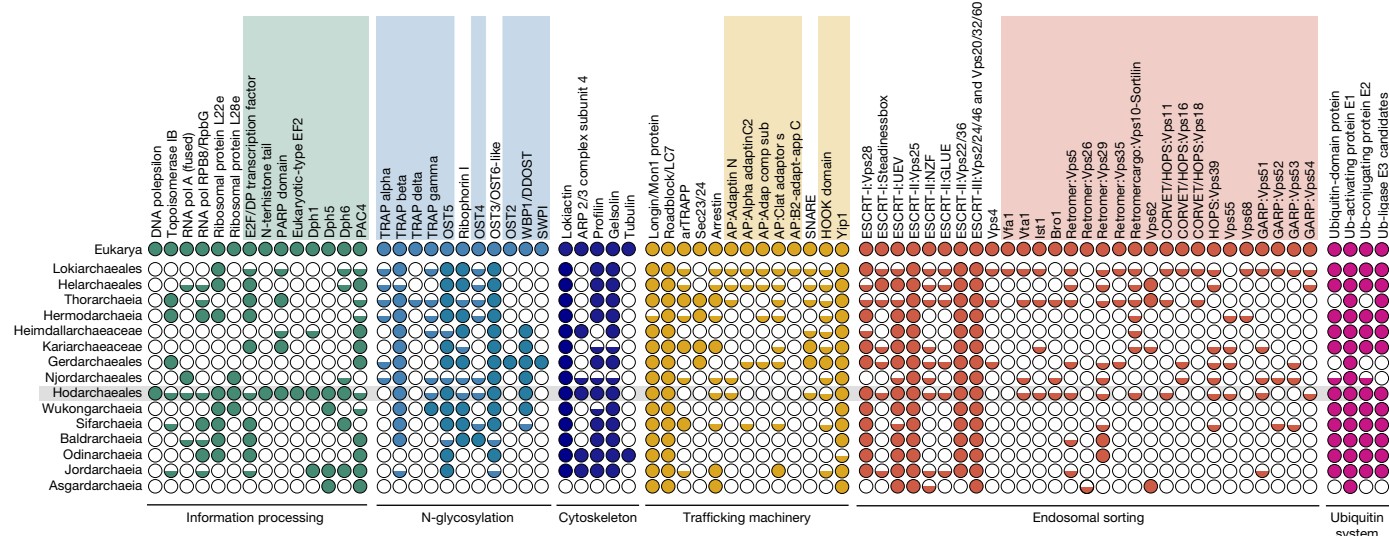

**Fig. 3 | Eukaryotic signature proteins in Asgard archaea.** Distribution of ESP homologues in Asgard archaea grouped by function. Shaded rectangles above the protein names indicate ESPs newly identified as part of this study. Predicted homologues are depicted by coloured circles: fully filled circles indicate that we detected homologues in at least half of the representative genomes of the clade; half-filled circles indicate that we detected homologues in fewer than half of the representative genomes of the clade. Hodarchaeales ESP homologues are highlighted against a grey background. Accession numbers are available in Supplementary Table 3.

encode relatively complex machineries for the N-linked glycosylation and translocation of proteins (Extended Data Fig. 2).

## Membrane-trafficking homologues

Intracellular vesicular transport represents a key process that emerged during eukaryogenesis. Previous studies have reported that Asgard archaeal genomes encode homologues of eukaryotic proteins comprising various intracellular vesicular trafficking and secretion machineries. These include the endosomal sorting complexes required for transport (ESCRT), transport protein particle (TRAPP) and coat protein complex II (COPII) vesicle coatomer protein complexes[2,3]. Furthermore, as much as 2% of the genes of Asgard archaeal genomes encode small GTPase homologues. These comprise a broad family of eukaryotic proteins encompassing the Ras, Rab, Arf, Rho and Ran subfamilies, which are broadly implicated in budding, transport, docking and fusion of vesicles in eukaryotic cells[2,3,33]. Here we report the identification of Asgard archaeal homologues of subunits of additional vesicular trafficking complexes (Fig. 3, Extended Data Fig. 3 and Supplementary Table 3). Notably, we found putative homologues of all four subunits comprising eukaryotic adaptor proteins and coatomer protein (COPI) complexes. In eukaryotic cells, these complexes are involved in the formation of clathrin-coated pits and vesicles responsible for packaging and sorting cargo for transport through the secretory and endocytic pathways[34]. They are composed of two large subunits, belonging to the β-family and γ-family, a medium μ-subunit and a small σ-subunit. We found homologues of all functional domains constituting these subunits, albeit sparsely distributed (Extended Data Fig. 3 and Supplementary Information). Additionally, we found homologues of several protein complexes involved in eukaryotic endosomal sorting such as the retromer, the homotypic fusion and protein sorting (HOPS), class C core vacuole/endosome tethering (CORVET) and the Golgi-associated retrograde protein (GARP) complexes (Fig. 3, red shading). Retromer is a coat-like complex associated with endosome-to-Golgi retrograde traffic[35], and we detected four out of its five subunits in Asgard archaeal MAGs. One of these subunits is Vps5-BAR, which in Thorarchaeia is often fused to Vps28, a subunit of the ESCRT-I subcomplex. This finding implicated a functional link between BAR domain proteins and the thorarchaeial ESCRT complex. The GARP complex is a multisubunit tethering

complex located at the trans-Golgi network in eukaryotic cells, where it also functions to tether retrograde transport vesicles derived from endosomes[36], similar to the retromer complex. GARP comprises four subunits, three of which we detected in Asgard archaeal genomes, with a sparse and punctuated distribution. Functioning in the opposite direction from the retromer and GARP complexes are the CORVET and HOPS complexes[37]. Endosomal fusion and autophagy in eukaryotic cells depend on them and they share four core subunits, three of which were found in Asgard archaea in addition to one of the HOPS-specific subunits.

Finally, although numerous components of the ESCRT-I, ESCRT-II and ESCRT-III systems have been previously detected in Asgard archaea[2,3,38], we report here the identification of Asgard archaeal homologues for the ESCRT-III regulators Vfa1, Vta1, Ist1 and Bro1.

## Ancestral Asgard archaea proteomes

The analysis of Asgard archaeal genome data obtained through metagenomics, combined with the insights derived from cytological observations of the first two cultured Asgard archaea 'Candidatus Prometheoarchaeum syntrophicum'[39] and 'Candidatus Lokiarchaeum ossiferum'[40], have generated new hypotheses about the nature of the archaeal ancestor of eukaryotes[39,41,42]. However, these theories are mostly based on a limited number of features displayed by a single or a few Asgard archaeal lineages. Although informative, features of present-day Asgard archaea do not necessarily resemble those of LAECA, as these are potentially separated by more than 2 billion years of evolution[43]. Furthermore, Asgard archaeal classes, and even orders, display a highly variable genome content with respect to ESPs and predicted metabolic features[39,42,44–46], which indicate a complex evolutionary history of those traits. In light of these considerations, we inferred ancestral features of LAECA by using a ML evolutionary framework. We used a probabilistic gene-tree species-tree reconciliation approach in combination with the extended taxonomic sampling of Asgard archaeal genomes to reconstruct the evolutionary history of homologous gene families and ancestral gene content across the Asgard archaeal species tree. For this, we inferred ML phylogenetic trees of all 17,200 protein families encoded across 181 archaeal genomes, including representatives from Asgard and TACK archaea and from Euryarchaea

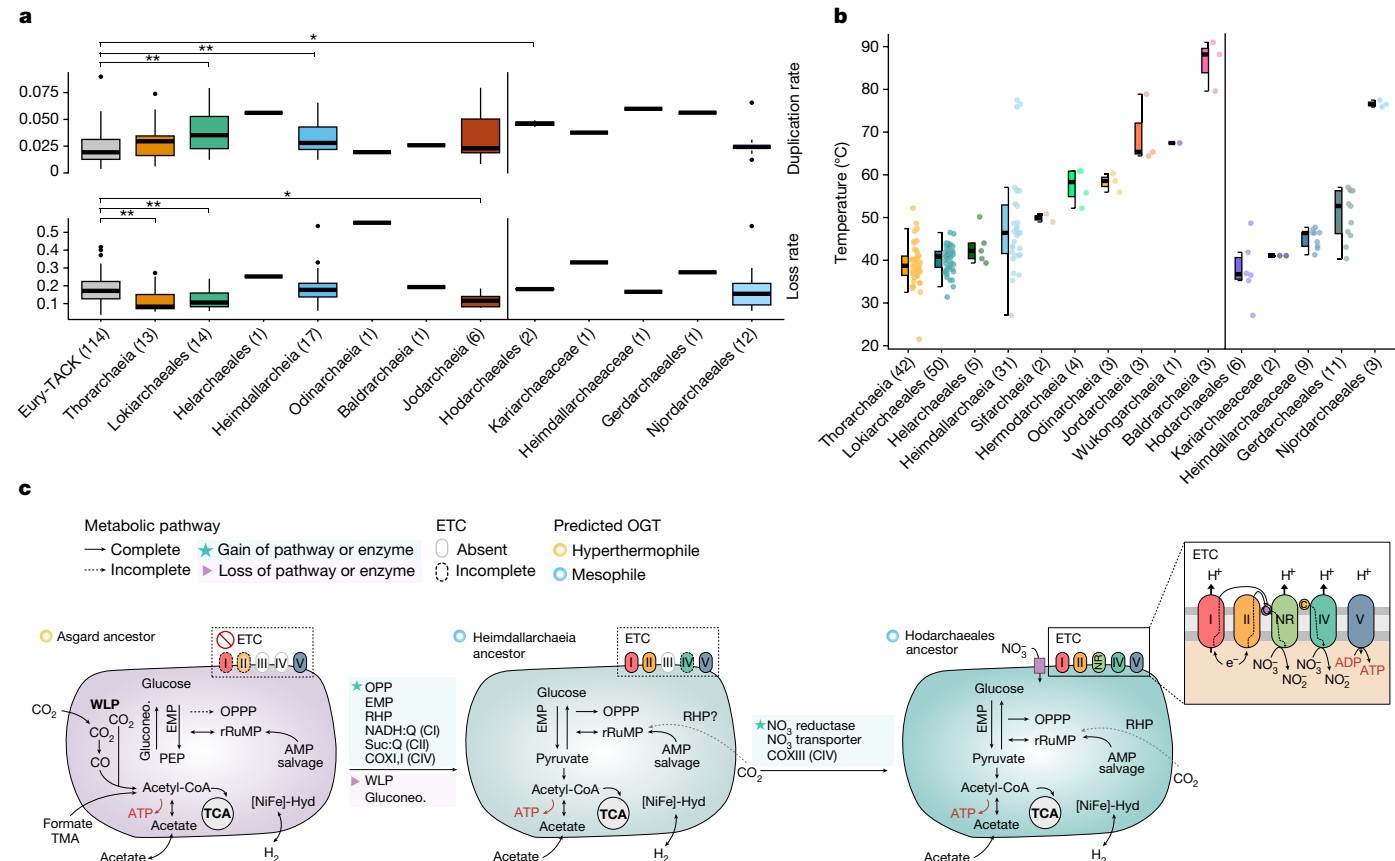

**Fig. 4 | Genome dynamics, OGT predictions and metabolic reconstruction of Asgard ancestors. a**, Duplication and loss rates inferred for Asgard archaeal ancestors, normalized by proteome size. *P* values given for each two-sided Wilcoxon-test against the median values of TACK and Euryarchaea (Eury-TACK) ancestors, where *$P \le 0.05$, **$P \le 0.01$ and ***$P \le 0.001$. No corrections were done for multiple comparisons. **b**, OGT predictions predicted by genomic features. Right, OGTs within Heimdallarchaeia. Actual values are available in Supplementary Table 5. In **a** and **b**, boxplots are represented as a central line denoting the median value, a coloured box containing the first and third quartiles of the dataset, and whiskers representing the lowest and highest values within 1.5 times the interquartile range, and sample sizes are shown within parentheses on the axis labels. **c**, We predict that the LAsCA transitioned from a hyperthermophilic fermentative lifestyle to a mesophilic mixotroph lifestyle. The LAsCA probably encoded gluconeogenic (Gluconeo.) pathways through the reverse EMP gluconeogenic pathway and through fructose

1,6-bisphosphate aldolase/phosphatase (FBP A/P). The major energy-conserving step in the early Asgard ancestors could have been the ATP synthesis by fermentation of small organic molecules (acetate, formate or formaldehyde). The reverse ribulose monophosphate pathway (rRuMP) was a key pathway in the LAsCA for the generation of reducing power. The WLP appeared only present in the LAsCA. The tricarboxylic acid (TCA) cycle is predicted complete in all three ancestors, the Hodarchaeales common ancestor encoding the most complete ETC, and probably used nitrate as a terminal electron acceptor. Membrane-associated ATP biosynthesis coupled to the oxidation of NADH and succinate and reduction of nitrate could have been present in the LAECA. c, cupredoxin; NR, nitrate reductase; OPPP, oxidative pentose phosphate pathway; PEP, phosphoenolpyruvate; PRK: phosphoribulokinase; Q, quinone; RHP, reductive hexulose-phosphate; RuBisCO, ribulose-1,5-bisphosphate carboxylase/oxygenase; TMA, trimethylamine.

clades. Of note, missing genes and potential contaminations in MAGs will be regarded as recent gene loss and gain events in our ancestral reconstruction analyses. Therefore, the use of incomplete MAGs with low contamination levels is unlikely to affect the inferred gene content of the deep archaeal ancestors that were reconstructed in the current study (Supplementary Information).

We first compared the distributions of estimated ancestral proteome sizes and the numbers of inferred gene duplications, losses and gains (that is, horizontal gene transfers and originations) in all archaeal ancestral nodes (Supplementary Fig. 12). Heimdallarchaeia (in particular the ancestor of Hodarchaeales) and Lokiarchaeia ancestors displayed significantly higher gene duplication rates compared with TACK and Euryarchaea ancestors (Fig. 4a). In addition, most Asgard archaeal ancestors displayed gene loss rates comparable with other archaea, with the exception of Thorarchaeia, Lokiarchaeales and Jordarchaeia, which showed significantly lower rates of loss. In agreement with the observed evolutionary genome dynamics, predicted proteome sizes of most Asgard archaea ancestors were significantly larger than other

archaeal ancestors ($P < 0.001$), with Lokiarchaeia ancestors displaying the largest estimated proteome size (Supplementary Fig. 13). Similarly, the Hodarchaeales ancestor had an estimated proteome size of 4,053 proteins compared with 3,134 for the last Asgard archaea common ancestor (LAsCA), which reflected the high duplication and low loss rates in that clade. The streamlined genome content of the Odinar-chaeia ancestor represents an exception to the general trend of genome expansion across Asgard archaea and possibly reflects an adaptation to high temperatures (Fig. 4b)[47].

## Ancestral features of the LAECA

Using the above-described approach, we reconstructed the ancestral metabolic and physiological properties across the Asgard archaeal species tree, including the proposed closest archaeal relatives of eukaryotes, the Hodarchaeales. We inferred that the LAsCA was a chemolithotroph that required the synthesis of organic building blocks through the Wood–Ljungdahl pathway (WLP) (Fig. 4c and

Supplementary Information), for which we inferred the presence of key enzymes, including carbon monoxide dehydrogenase/acetyl-CoA synthase and the formylmethanofuran dehydrogenase. In addition, our analyses revealed that the last common ancestors of individual Asgard archaeal classes either had the genetic potential to switch between autotrophy and heterotrophy (Lokiarchaeia, Thorarchaeia, Jordarchaeia and Baldrarchaeia) or a predominantly heterotrophic fermentative (Odinarchaeia and Heimdallarchaeia) lifestyle (Fig. 4c and Supplementary Information). Specifically, we observed that the WLP was lost before the last common ancestor of Heimdallarchaeia (and therefore before the emergence of LAECA), which indicated that the LAECA was a heterotrophic fermenter (Supplementary Table 4).

Furthermore, we inferred that the central carbon metabolism of Heimdallarchaeia (including Hodarchaeales) included the Embden–Meyerhof–Parnas (EMP) pathway and a partial oxidative pentose phosphate pathway—both considered core modules of present-day eukaryotic central carbon metabolism. Although the enzymes of these pathways in Asgard archaea do not share a common evolutionary origin with those of eukaryotes, this inference suggests that the LAECA had a similar central carbon metabolism compared to modern eukaryotes (Supplementary Figs. 14 and 15).

In addition, our analyses support the idea that the last common ancestor of Heimdallarchaeia contained several components of the electron transport chain (ETC)[42]. We inferred that the last common ancestor of Hodarchaeales probably contained CI, CII, CIV and a nitrate reductase complex (NarGHIJ), which indicated that nitrate might have been used as a terminal electron acceptor to perform anaerobic respiration. As such, the last Hodarchaeales common ancestor probably generated ATP using an ETC whereby electrons from NADH and succinate were transferred through a series of membrane-associated complexes with quinones and cupredoxins as electron carriers to ultimately reduce nitrate[48].

As indicated above, a substantial fraction of the currently sampled Asgard archaea diversity originated from geothermal or hydrothermal environments. Using an algorithm based on genome-derived features, we confirmed that (most) Njordarchaeales, Baldrarchaeia and Jordarchaeia are hyperthermophiles, Odinarchaeia are thermophiles, and Lokiarchaeia and Thorarchaeia are mesophiles (Fig. 4b and Supplementary Table 5). Whereas Heimdallarchaeia seemed to contain both mesophiles and thermophiles, we inferred a mesophilic physiology for Hodarchaeales, obtaining the lowest predicted optimal growth temperatures (OGTs) among all Asgard archaea (median = 36.7 °C). Asgard archaeal hyperthermophiles contained reverse gyrase, a topoisomerase that is typically encoded by hyperthermophilic prokaryotes. We inferred that a reverse gyrase was possibly present in the LAsCA and that it was subsequently lost in all heimdallarchaeial orders except for Njordarchaeales. This observation would be compatible with a scenario in which Asgard archaea have a hyperthermophilic ancestry, but in which eukaryotes evolved from an Asgard archaea lineage that had adapted to mesophilic growth temperatures.

## Discussion

Beyond genomic exploration, several studies have started to uncover important physiological, cytological and ecological aspects of Asgard archaea[38,39,49–51]. Yet, although such insights are relevant, the cellular and physiological characteristics of present-day Asgard archaea will probably not resemble those of the LAECA. Therefore, inferences about the identity and nature of the LAECA and the process of eukaryogenesis should be made within an evolutionary context. We used an evolutionary framework to analyse an expanded Asgard archaeal genomic diversity comprising 11 clades of high taxonomic rank. We also performed comprehensive phylogenomic analyses involving the evaluation of distinct marker protein datasets and systematic assessments of suspected phylogenetic artefacts and state-of-the-art models of evolution.

As a result, we identified Hodarchaeales, an order-level clade within the Heimdallarchaeia, as the closest relatives of eukaryotes. Evidently, phylogenomic analyses that aim to pinpoint the phylogenetic position of eukaryotes in the tree of life are challenging, and our results stress the importance of testing for possible sources of bias that affect phylogenomic reconstructions, as was recently reviewed[52]. The implementation of a probabilistic gene tree or species tree reconciliation approach enabled us to infer the evolutionary dynamics and ancestral content across the archaeal species tree, providing several new insights into the Asgard archaeal roots of eukaryotes. Altogether, our results revealed a picture in which the Asgard archaeal ancestor of eukaryotes had, compared with other archaea, a relatively large genome that resulted mainly from more numerous gene duplication and fewer gene loss events. It is tempting to speculate that the increased gene duplication rates observed in our analyses represent an ancestral feature of the LAECA and that it remained the predominant mode of genome evolution during the early stages of eukaryogenesis. We also inferred that the duplicated gene content of the LAECA included several protein families involved in cytoskeletal and membrane-trafficking functions, including, among others, actin homologues, ESCRT complex subunits and small GTPase homologues. Our findings complement those of another study[53] reporting that eukaryotic proteins with an Asgard archaeal provenance, as opposed to those inherited from the mitochondrial symbiont, duplicated the most during eukaryogenesis, particularly proteins of cytoskeletal and membrane-trafficking families.

Beyond genome dynamics, our analyses of inferred ancestral genome content across the Asgard archaeal species tree indicated that although Asgard archaea probably had a thermophilic ancestry, the lineage from which eukaryotes evolved was adapted to mesophilic conditions. This finding is compatible with a generally assumed mesophilic ancestry of eukaryotes. Furthermore, we inferred that the LAECA had the genetic potential to support a heterotrophic lifestyle and may have been able to conserve energy through nitrate respiration. In addition, on the basis of taxonomic distribution and evolutionary history of ESPs, we showed that complex pathways involved in protein targeting and membrane trafficking and in genome maintenance and expression in eukaryotes were inherited from their Asgard archaeal ancestor. Of note, we identified additional Asgard archaeal homologues of components of eukaryotic vesicular trafficking complexes. Of these, some Asgard archaeal proteins displayed sequence similarity to proteins that, in eukaryotes, are part of the clathrin adaptor protein complexes and of the COPI complex. These complexes are particularly interesting because they are involved in the biogenesis of vesicles responsible for sorting cargo and subsequent transport through the secretory and endocytic pathways[34]. Altogether, these results further suggest the potential for membrane deformation, and possibly trafficking, in Asgard archaea. The ability to deform membranes was recently shown in two papers reporting the first cultivated Lokiarchaeia lineages, 'Ca. Promethearchaeum syntrophicum strain MK-D1'[39] and 'Ca. Lokiarchaeum ossiferum'[40], the cells of which both displayed distinct morphological complexity, including long and often branching protrusions facilitated by a dynamic actin cytoskeleton. Thus far no[39], or only limited[40], visible endomembrane structures have been observed in these first cultured representatives of Asgard archaea. However, it is important to restate here that, being separated by some 2 billion years of evolution, the cellular features of present-day Asgard archaeal lineages do not necessarily resemble those of the LAECA. Furthermore, given the disparity of the distribution patterns of membrane-trafficking homologues in Asgard archaea, it will be crucial to isolate representatives of classes other than Lokiarchaeia and to study their cell biology features and potential for endomembrane biogenesis. Of particular interest would be members of the Heimdallarchaeia and specifically Hodarchaeales, as the currently identified closest relatives of eukaryotes, as well as Thorarchaeia lineages, which seem to generally contain a particularly rich suite of homologues of eukaryotic membrane-trafficking proteins.

Our work phylogenetically places eukaryotes as a nested clade within the currently identified Asgard archaeal diversity, and we inferred ancestral genomic content across the Asgard archaea. These results provide insights into the identity and nature of the Asgard archaeal ancestor of eukaryotes, guiding future studies that aim to uncover new pieces of the eukaryogenesis puzzle.

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

## Methods

### Sample collection, sequencing, assembly and binning

We sampled aquatic sediments from 11 geographically distant sites: Guaymas Basin (Mexico); Lau Basin (Eastern Lau Spreading Center and Valu Fa Ridge, south-west Pacific Ocean); Hydrate Ridge (offshore of Oregon, USA); Aarhus Bay (Denmark); Radiata Pool (New Zealand); Taketomi Island Vent (Japan); the White Oak River estuary (USA); and Tibet Plateau and Tengchong (China) (Supplementary Table 1).

**Sampling permissions.** The following sampling permits were used: Guaymas Basin (DAPA/2/251108, DAPA/2/131109/3958 and CONAPES-CA); ABE and Mariner field (TN-002-2015, Kingdom of Tonga); and Radiata pool (77982-RES, Department of Conservation (New Zealand)). No permits were needed for obtaining any of the other samples described in this study. Additional information regarding sampling years and responsible scientists are available in Supplementary Table 1.

**Tibet Plateau and Yunnan Province.** For Jordarchaeia JZB50, QC4B49, QZMA23B3, QZMA2B5 and QZMA3B5, samples from hot spring sediment were collected from Tibet Plateau and Yunnan Province (China) in 2016. The microbial community compositions have been described and previously reported[54,55]. Samples were collected from the hot spring pools using a sterile iron spoon into 50 ml sterile plastic tubes, then transported to the laboratory on dry ice and stored at −80 °C until DNA extraction. The genomic DNA of the sediment samples was extracted using a FastDNA Spin Kit for Soil (MP Biomedicals) according to the manufacturer's instructions. The obtained genomic DNA was purified for library construction and sequenced on an Illumina HiSeq2500 platform (2× 150 bp). The raw reads were filtered to remove Illumina adapters, PhiX and other Illumina trace contaminants using BBTools (v.38.79), and low-quality bases and reads were removed using Sickle (v.1.33; https://github.com/najoshi/sickle). The filtered reads were assembled using metaSPAdes (v.3.10.1) with a kmer set of "21, 33, 55, 77, 99, 127". The filtered reads were mapped to the corresponding assembled scaffolds using bowtie2 (v.2.3.5.1)[56]. The coverage of a given scaffold was calculated using the command of jgi_summarize_bam_contig_depths in MetaBAT (v.2.12.1)[57]. For each sample, scaffolds with a minimum length of 2.5 kbp were binned into genome bins using MetaBAT (v.2.12.1), with both tetranucleotide frequencies and scaffold coverage information considered. The clustering of scaffolds from the bins and the unbinned scaffolds was visualized using ESOM with a minimum window length of 2.5 kbp and a maximum window length of 5 kbp, as previously described[58]. Misplaced scaffolds were removed from bins, and unbinned scaffolds for which segments were placed within the bin areas of ESOMs were added to the corresponding bins. Scaffolds with a minimum length of 1 kbp were uploaded to ggKbase (http://ggkbase.berkeley.edu/). The ESOM-curated bins were further evaluated based on consistency of GC content, coverage and taxonomic information, and scaffolds identified with abnormal information were removed. The ggKbase genome bins were individually curated to fix local assembly errors using ra2.py[59].

**ABE and Mariner hydrothermal vent fields.** For Heimdallarchaeia A173, A3132 and M288, and Thorarchaeia A361, A381 and A399, hydrothermal vent deposits were collected from ABE (ABE 1, 176° 15.48′ W, 21° 26.68′ S, 2,142 m; ABE 3, 176° 15.59′ W, 21° 26.95′ S, 2,131 m) and Mariner (176° 36.07′ W, 22° 10.81′ S, 1,914 m) vent fields along the Eastern Lau Spreading Center in April and May of 2015 during the RR1507 Expedition on the RV Roger Revelle. Sample collection and processing were done as previously described[60]. DNA was extracted from homogenized rock slurries using a DNeasy PowerSoil kit (Qiagen) as per the manufacturer's instructions. Samples were prepared for sequencing on an Illumina HiSeq 3000 using Nextera DNA Library Prep kits (Illumina), and metagenomes (2× 150 bp) were sequenced at the Oregon State University Center for Genome Research and Computing. Trimmomatic (v.0.36)[61] was used to trim low-quality regions and adapter sequences from raw reads (parameters: ILLUMINACLIP:TruSeq3-PE-2.fa:2:30:10, LEADING:20, SLIDINGWINDOW:4:20, MINLEN:50). Clean paired reads were then interleaved using the khmer software package[62]. Interleaved and unpaired reads were assembled using MEGAHIT (v.1.1.1-2-g02102e1) (--k-min 31, --k-max 151, --k-step 20, --min-contig-len 1000)[63,64]. Trimmed reads were mapped back to the contigs to determine read coverage using Bowtie 2 (v.2.2.9)[56,65] and SAMtools (v.1.3.1)[66]. Binning was performed using MetaBAT (v.0.32.4)[57] and tetranucleotide frequency and read coverage. Bin completion and contamination were estimated using CheckM (v.1.0.7)[67].

**Aarhus Bay.** For Lokiarchaeia ABR01, ABR02, ABR03, ABR04, ABR05, ABR06, ABR08, ABR11, ABR13 and ABR15, Thorarchaeia ABR09 and ABR10 and Heimdallarchaeia ABR14 and ABR16, MAGs were obtained as previously described[29].

**White Oak River.** For Sifarchaeia WORA1, Hermodarchaeia WORB2, Heimdallarchaeia WORE3, Lokiarchaeia WORB4 and WORC5, and Thorarchaeia WORH6, sampling, DNA extraction, sequencing library preparation and sequencing methods were performed as previously described[68]. Published assemblies and raw reads for the samples WOR-1-36_30 (National Center for Biotechnology Information (NCBI) BioSample identifier SAMN06268458; Joint Genome Institute (JGI) identifier Gp0056175), WOR-1-52-54 (SAMN06268416; Gp0059784), WOR-3-24_28 (SAMN06268417; Gp0059785) were downloaded from the JGI. Short reads were trimmed using Trimmomatic (v.0.33)[61] (PE ILLUMINACLIP:2:30:10 SLIDINGWINDOW:4:15 MILEN:100). Contigs shorter than 1,000 bp were excluded from the assembly using SeqTK (v.1.0r75) (https://github.com/lh3/seqtk). Each assembly was binned using CONCOCT (v.0.4.1)[69] and coverage information from the three datasets, and Asgard bins were subsequently identified based on phylogenies of concatenated ribosomal proteins[3]. Identified Asgard MAGs were used together with publicly available Asgard genomes to recruit trimmed reads originated from Asgard genomes using CLARK (v.1.2.3) with the -m 0 option[70]. For each dataset, recruited Asgard reads were independently assembled using SPAdes[71] and IDBA-UD[72] and further binned using CONCOCT, using a minimum contig length of 1,000 bp. Bins with higher completeness and lower contamination values as predicted by miComplete (v.1.00)[73] were selected and manually curated using mmgenome (v.0.7.1)[74,75] using the coverage information, paired-reads linkage, composition and marker genes information. The samples and assembly method used for each final MAG were as follows: Sifarchaeia WORA1 (WOR-1-52-54; spades); Hermodarchaeia WORB2 (WOR-1-52-54; IDBA-UD); Heimdallarchaeia WORE3 (WOR-3-24_28; spades); Lokiarchaeia WORB4 and WORC5 (WOR-1-36_30; IDBA-UD); and Thorarchaeia WORH6 (WOR-1-36_30; spades).

**Radiata Pool hot springs.** For Jordarchaeia RPD1 and RPF2, and Odinarchaeia RPA3, information about the location of the hot spring sediments from Radiata Pool, sampling and DNA extraction procedures has been previously reported[3]. Short paired-end Illumina reads were generated and preprocessed using Scythe (https://github.com/vsbuffalo/scythe) and Sickle (https://github.com/najoshi/sickle) to remove adapters and low-quality reads. Reads were subsequently assembled with IDBA-UD 1.1.3 (--maxk 124). The Jordarchaeia RPF2 MAG was generated by binning contigs according to their tetranucleotide frequencies using esomWrapper.pl (https://github.com/tetramerFreqs/Binning) with a minimum contig length of 5,000 bp and a window size of 10 kbp. ESOM maps were manually delineated using the Databionic ESOM viewer (http://databionic-esom.sourceforge.net/). Jordarchaeia RPD1 and Odinarchaeia RPA3 were binned following the previously described[29] methodology, but re-assembling the recruited reads only with IDBA-UD (--maxk 124)[72].

**Guaymas Basin.** For Asgardarchaeia GBS01, Baldrarchaeia GBS02, GBS03, and GBS04, Jordarchaeia GBS05, GBS06 and GBS07, Heimdallarchaeia GBS08, GBS09, GBS10, GBS11, GBS15, GBS16, GBS17, GBS18, GBS19, GBS20, GBS21, GBS22, GBS23, GBS24, GBS25, GBS26 and TNS08, Lokiarchaeia GBS14, and Thorarchaeia GBS28, GBS29, GBS33 and GBS34, MAGs were obtained as previously described[76]. For Heimdallarchaeia GBS09, the MAG was obtained as previously described[77].

**South Hydrate Ridge.** For Heimdallarchaeia GBS11, samples were made available by the Gulf Coast Repository (GCR) and were collected on the Ocean drilling Program (ODP) Leg 204 at site 1244 (44° 35.17 N, 125° 7.19 W) on 14 July 2002 (hole C and core 2). The ODP site is found at a water depth of 890 m on the eastern side of the South Hydrate Ridge on the Cascadia Margin. This site has been well characterized physically and geochemically[78]. Furthermore, the microbial community structure has been surveyed using 16S rRNA gene sequencing[79,80]. Two sediment samples, designated DCO-2-5 (sample identifier 1489929) and DCO-2-7 (sample identifier 1489924), were collected at a sediment depth of 12.40 and 14.96 m below the seafloor, respectively, and stored at −80 °C at GCR. A total of 10 g of each of the two sediment samples was used to extract DNA using a MoBio DNA PowerSoil Total kit. A total of 100 ng DNA was used to prepare sequencing libraries that were 150 bp paired-end sequenced at the Marine Biological Laboratory (Woods Hole, MA, USA) on an Illumina MiSeq sequencer. Adaptors and DNA spike-ins were removed from the forward and reverse reads using cutadapt (v.1.12)[81]. Afterwards, reads were interleaved using interleave_fasta.py (https://github.com/jorvis/biocode/blob/master/fasta/interleave_fasta.py) and further trimmed using Sickle with default settings (Fass JN) (https://github.com/najoshi/sickle). Metagenomic reads from both samples were co-assembled using IDBA-UD with the following parameters: --pre_correction, --mink 75, --maxk 105, --step 10, --seed_kmer 55 (ref. 72). Metagenomic binning was performed on scaffolds with a length of >3,000 bp using ESOM, including a total of 4,939 scaffolds with a length of 30,693,002 bp[58,72]. CheckM (v.1.0.5) was used to evaluate the accuracy of the binning approach by determining the percentage of completeness and contamination[67].

### Exploration of phylogenetic diversity in Asgard archaeal assemblies and MAGs

To assess the presence of potential Asgard-related lineages in our assemblies, we reconstructed a phylogeny of ribosomal proteins encoded in a conserved RP15 gene cluster[82]. As the in-group, we used all MAGs presented in this study, plus all genomes classified as Asgard archaea in the NCBI database as of 25 June 2021, plus those classified as 'archaeon' corresponding to Hermodarchaeia (GCA_016550385.1, GCA_016550395.1, GCA_016550405.1, GCA_016550415.1, GCA_016550425.1, GCA_016550485.1, GCA_016550495.1 and GCA_016550505.1), and all Asgard archaeal MAGs released in previous study[19]. To obtain an adequate outgroup dataset, we downloaded all archaeal genomes from the Genome Taxonomy Database[83], release 89, and selected one genome sequence per species-level cluster as previously defined (https://data.gtdb.ecogenomic.org/releases/release89/89.0/sp_clusters_r89.tsv). We then selected a set of 216 genomes classified as Bathyarchaeia, Nitrososphaeria and Thermoprotei, and used them as the outgroup. Genes were detected and individually aligned and trimmed as previously described[3]. Ribosomal protein sequences were selected if they were encoded in a contig containing at least 5 out of the 15 ribosomal protein genes. ModelFinder[84] was run as implemented in IQ-TREE (v.2.0-rc2) to identify the best model among all combinations of the LG, WAG, JTT and Q.pfam models, as well as their corresponding mixture models by adding +C20, +C40 and +C60, and the additional mixture models LG4M, LG4X, UL2 and UL3, with rate heterogeneity (none, +R4 and +G4) and frequency parameters (none, +F). A PMSF approximation[85] of the

chosen model (WAG+C60+R4+F) was then used for a final reconstruction using 100 nonparametric bootstrap pseudoreplicates for branch statistical support. The obtained tree revealed a broad genomic diversity of Asgard lineages (Fig. 1).

### Gene prediction

Gene prediction was performed using Prokka (v.1.12)[86] (prokka --kingdom Archaea --norrna --notrna). rRNA genes and tRNA genes were predicted using Barrnap (https://github.com/tseemann/barrnap) and tRNAscan-SE[87,88], respectively.

### OGT prediction

OGT values were predicted for the genomes presented here based on genomic and proteomic features[89] (Supplementary Information). As rRNA nucleotide compositions are used in this method, only genomes with predicted rRNAs were analysed.

### Identification of homologous protein families

All-versus-all similarity searches of all predicted proteins from the A64 taxon selection (64 Asgard, 76 TACK, 43 Euryarchaea and 41 DPANN archaea; Supplementary Table 2) were performed using diamond[90] BLASTp (--more-sensitive --evalue 0.0001 --max-target-seqs 0 --outfmt 6). The file generated was used to cluster protein sequences into homologous families using SiLiX (v.1.2.10)[91] followed by Hifix (v.1.0.6)[92]. The identity and overlap parameters required by Silix were set to 0.2 and 0.7, respectively, after inspecting a wide range of values (--ident [0.15,0.4] and --overlap [0.55–0.9], with increments of 0.05) and selecting the values that maximized the number of clusters containing at least 80% of the taxa.

### Functional annotation of homologous protein families

Protein families, excluding singletons, were aligned using mafft-linsi (v.7.402)[93] and converted into HHsearch format (.hhm) profiles using HHblits (v.3.0.3)[94]. Profile–profile searches were subsequently performed against a database containing profiles from EggNOG (v.4.5)[95], arCOGs[96] and Pfam databases[97] that had been previously converted to the hhm format using HHblits (v.3.0.3)[94].

### Detailed analysis of ESPs

In-depth analysis of potential ESPs involved a combination of automatic screens and manual curation. We first manually searched for homologues of previously described ESPs[2,3,38] by using a variety of sequence similarity approaches such as BLAST, HMMer tools, profile–profile searches using HHblits, combined with phylogenetic inferences, and, in some cases, the Phyre2 structure homology search engine[94,98,99]. We did not use fixed cutoffs, as the e-value between homologues will vary depending on the protein investigated, hence the need for manual examination of potential homologues and a combination of lines of evidence.

In addition, to identify potential new ESPs, we first used our profile–profile searches against EggNOG and manually investigated Asgard orthologous groups that had a best hit to a eukaryotic-specific EggNOG cluster. We also extracted Pfam domains for which the taxonomic distribution are exclusive to eukaryotes as per Pfam (v.32), and investigated cases in which they represented the best domain hit in Asgard archaea sequences identified by HMMscan. Finally, we manually investigated dozens of proteins known to be involved in key eukaryotic functions based on our knowledge and literature searches. In Fig. 2, we are only reporting cases based on the strict cutoff that the diagnostic HMM profile had the best score among all profiles detected for a protein. An exception was made for the ESCRT domain Vps28, Steadiness box, UEV, Vps25, NZF, GLUE and Vps22 domains, which are usually found in combination with other protein domains and thus do not necessarily represent the best scoring domain in a protein even if they represent true homologues.

## Phylogenetic analyses of concatenated proteins for species tree inference

Two sets of phylogenetic markers were used to infer the species tree. The first one (RP56) is based on a previously published dataset of 56 ribosomal proteins used to place the first assembled Asgard genomes[3]. The second one (NM54, for new markers) corresponds to 54 proteins extracted from a set of 200 markers previously identified as core archaeal proteins that can be used to confidently infer the tree of archaea[100]. These 54 markers were selected because they were found in at least one-third of representatives of each of the 11 Asgard clades, as well as in 10 out of 14 eukaryotes, and were inherited from archaea in eukaryotes.

We initially assembled a RP56 dataset for a phylogenetically diverse set of 222 archaeal and 14 eukaryotic taxa. These included all 11 Asgard archaea MAGs and genomes available at the NCBI as of 12 May 2017, as well as the 53 most diverse new MAGs from this work (out of 63). We gathered orthologues of these genes from all proteomes by using sequences from the previously published alignment[3,100] as queries for BLASTp. For each marker, the best BLAST hit from each proteome was added to the dataset. For the first iteration, each dataset was aligned using mafft-linsi[101] and ambiguously aligned positions were trimmed using BMGE (-m BLOSUM30)[102]. All 56 trimmed ribosomal protein alignments were concatenated into a RP56-A64 supermatrix (236 taxa including 64 Asgard archaea, 6,332 amino acid positions). Once this taxon set was gathered, we identified homologues of the NM54 gene set as described above, thus generating supermatrix NM54-A64 (236 taxa, 14,847 amino acid positions).

We carried out a large number of phylogenomic analyses on variations of these two RP56-A64 and NM54-A64 datasets with different phylogenetic algorithms. Notably, preparing these datasets must be done with great care and is therefore time-consuming, and subsequent phylogenomic analyses generally require an enormous amount of computational running time. However, the rapid expansion of available Asgard archaeal MAGs, notably in a previous publication[4], urged us to update and re-run many of the computationally demanding analyses. As some of the work that was based on a more restrained taxon sampling is still deemed valuable, such as some of the Bayesian phylogenomic analyses and ancestral genome content reconstructions, we retained these in the current study.

An updated Asgard archaeal genomic sequence dataset was constructed by including all 230 Asgard archaeal MAGs and genomes available at the NCBI database as of 12 May 2021, as well as 63 new MAGs described in the current work. All 56 trimmed ribosomal protein alignments were concatenated into an RP56-A293 supermatrix (465 taxa including 293 Asgard archaea, 7,112 amino acid positions), which was used to infer a preliminary phylogeny using FastTree (v.2)[103] (Supplementary Fig. 16). Given the high computational demands of the subsequent analyses, we then used this phylogeny to select a subsample of Asgard archaea representatives. For this, we first removed the most incomplete MAGs encoding fewer than 19 ribosomal proteins (that is, one-third of the markers) in the matrix. We also used the preliminary phylogeny to subselect among closely related taxa: among taxa that were separated by branch lengths of <0.1, we only kept one representative. This led to a selection of 331 genomes, including 175 Asgard archaea, 41 DPANN, 43 Euryarchaea and 72 TACK representatives (RP56-A175 dataset). Out of these 175 Asgard archaea, 41 correspond to MAGs newly reported here. Once this taxon set was gathered, we identified homologues of the NM54 gene set as described above, thus generating supermatrix NM54-A172 (15,733 amino acid positions; three additional Asgard archaea were removed for having too few homologs of this gene set). All datasets and their composition are summarized in Supplementary Table 2.

To test for potential phylogenetic reconstruction artefacts, our datasets were subjected to several treatments. Supermatrices were recoded into four categories using the SR4 scheme[25]. The corresponding phylogenies were reconstructed using IQ-TREE (using a user-defined previously described model referred to as C60SR4 based on the implemented C60 model and modified to analyse the recoded data[3]) and Phylobayes (under the CAT+GTR model). We also used the estimated site rate output generated by IQ-TREE (-wsr) to classify sites into 10 categories, from the fastest to the slowest evolving, and we removed them in a stepwise fashion, removing from 10% to 90% of the data. Finally, we combined both approaches by applying SR4 recoding to the alignments obtained after each fast-site removal step. All phylogenetic analyses performed are summarized in Supplementary Table 2. See Supplementary Information for details and discussion.

## Analyses of individual proteins

For individual proteins of interest, we gathered homologues using various approaches depending on the level of conservation across taxa. To detect putative Asgard homologues of eukaryotic proteins, we used a combination of tools, including BLASTp[104] and the HMMer toolkit (http://hmmer.org/) if HMM profiles were available, and queried a local database containing our 240 archaeal representatives (including all Asgard predicted proteomes). We then investigated the Asgard candidates as following: (1) using them as seed for BLASTp searches against the nr database; (2) 3D modelling using Phyre2 and SwissModel when sequence similarity was low; (3) annotating them using Interproscan (v.5.25-64.0)[105], EggNOG mapper (v.0.12.7)[106], against the NOG database[106], and GhostKoala annotation server[107]; (4) annotating the archaeal orthologous cluster they belonged to using profile–profile annotation as described above. Eukaryotic homologues were gathered from the UniRef50 database[108]. Depending on the divergence between homologues, they were aligned using mafft-linsi and trimmed using TrimAl[109] (--automated1) or BMGE[102], or, in cases where we investigated a specific functional domain, we used the hmmalign tool from the HMMer package with the --trim flag to only keep and align the region corresponding to this domain. When divergence levels allowed, phylogenetic analyses were performed using IQ-TREE with model testing including the C-series mixture models (-mset option)[110]. Statistical support was evaluated using 1,000 ultrafast bootstrap replicates (for IQ-TREE)[109].

## Ancestral reconstruction

For the ancestral reconstruction analyses, only a subset of 181 taxa were included (64 Asgard, 74 TACK and 43 Euryarchaea; see Supplementary Table 2 for details). Protein families with more than three members were aligned and trimmed using mafft-linsi (v.7.402)[101] and trimAl (v.1.4.rev15) with the --gappyout option[109]. Tree distributions for individual protein families were estimated using IQ-TREE (v.1.6.5) (-bb 1000 -bnni -m TESTNEW -mset LG -madd LG+C10,LG+C20 -seed 12345 -wbtl -keep-ident)[111]. The species phylogeny together with the gene tree distributions were subsequently used to compute 100 gene–tree species tree reconciliations using ALEobserve (v.0.4) and ALEml_undated[112,113], including the fraction_missing option that accounts for incomplete genomes. The genome copy number was corrected to account for the extinction probability per cluster (https://github.com/maxemil/ALE/commit/136b78e). The missing fraction of the genome was calculated as 1 minus the completeness values (in fraction) as estimated by CheckM (v.1.0.5)[67] for each of the 181 taxa[67]. Protein families containing only one protein (singletons) were considered as originations at the corresponding leaf. The ancestral reconstruction of 5 protein families that included more than 2,000 proteins raised errors and could not be computed. The minimum threshold of the raw reconciliation frequencies for an event to be considered was set to 0.3 as commonly done[114–117] and recommended by the authors of ALE (G. Szöllősi, personal communication).

## Ancestral metabolic inferences

Metabolic reconstruction of the Asgard ancestors was based on the inference, annotation and copy number of genes in ancestral nodes.

The presence of a given gene was scored if its copy number in the ancestral nodes was above 0.3. A protein family was scored as 'maybe present' if the inferred copy number was between 0.1 and 0.3. The protein annotation of each of the clusters containing the ancestral nodes was manually verified for each of the enzymatic steps involved in the pathways, as detailed in Supplementary Table 4.

## Reporting summary

Further information on research design is available in the Nature Portfolio Reporting Summary linked to this article.

## Data availability

The MAGs reported in this study have been deposited at the DNA Data Bank of Japan, the European Molecular Biology Laboratory and GenBank. BioProject identifiers, BioSample identifiers and GenBank assembly accession numbers are provided in Supplementary Table 1. All raw data underlying phylogenomic analyses (raw and processed alignments and corresponding phylogenetic trees), and all predicted proteomes have been deposited into Figshare (https://doi.org/10.6084/m9.figshare.29436380).

## Code availability

Custom code used for data analysis is available at GitHub: https://github.com/laurajjeme/phylogenetics.

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

**Acknowledgements** We thank S. Köstlbacher, L. Hederstedt, A. Spang and A. J. Roger for discussions; staff at the Uppsala Multidisciplinary Center for Advanced Computational Science (UPPMAX) at Uppsala University and the Swedish National Infrastructure for Computing (SNIC) at the PDC Center for High-Performance Computing for providing computational resources; staff at the Japan Agency for Marine-Earth Science and Technology (JAMSTEC) for taking sediment samples from the Taketomi shallow submarine hydrothermal system; and the crew of the *RV Roger Revelle* for assisting with the sampling of the ABE and Mariner vent fields along the Eastern Lau Spreading Center during the RR1507 Expedition. The Ngāti Tahu–Ngāti Whaoa Runanga Trust is acknowledged as *mana whenua* of Radiata Pool and associated samples, and we thank them for their assistance in access and sampling of the Ngatamariki geothermal features. We thank the Kingdom of Tonga for access to the deep-sea hydrothermal vent sites along the ELSC. Sampling in the Eastern Lau Spreading Center and Guaymas Basin (Gulf of California) was supported by the US-National Science Foundation (NSF-OCE-1235432 to A.-L.R. and NSF-OCE-0647633 to A.P.T.). A subset of Guaymas sediments were sequenced by the US Department of Energy Joint Genome Institute, a DOE Office of Science User Facility under contract number DE-AC02-05CH11231 granted to N.D. We thank the captain and crew of *RV Aurora* for assistance during sampling at Aarhus Bay. Sampling at Aarhus Bay was supported by the VILLUM Experiment project "FISHing for the ancestors of the eukaryotic cell" (grant number 17621 to A.S. and K.U.K.). This work was supported by grants of the European Research Council (ERC Starting and Consolidator grants 310039 and 817834, respectively), the Swedish Research Council (VR grant 2015-04959), the Dutch Research Council (NWO-VICI grant VI.C.192.016), Marie Skłodowska-Curie ITN project SINGEK (H2020-MSCA-ITN-2015-675752) and the Wellcome Trust foundation (Collaborative award 203276/K/16/Z) awarded to T.J.G.E. L.E. was supported by a Marie Skłodowska-Curie IEF (grant 704263) and by funding from the European Research Council (ERC Starting grant 803151). T.N. was supported by JSPS KAKENHI JP19H05684 within JP19H05679. W.-J.L. was supported by the National Natural Science Foundation of China (grant number 91951205 and 92251302). D.T. was supported by the Swedish Research Council (International Postdoc grant 2018-06609). C.W.S. was supported by a Science for Life Laboratory postdoctoral fellowship (awarded to T.J.G.E.) and funding from the Swedish research council (Vetenskaprådet Starting grant 2020-05071 to C.W.S.). J.L. was supported by the Wenner-Gren Foundation (fellowship 2016-0072). J.H.S. was supported by a Marie Skłodowska-Curie IIF grant (331291). This work was also supported by the Moore-Simons Project on the Origin of the Eukaryotic Cell, Simons Foundation 73592LPI to T.J.G.E. and B.J.B. (https://doi.org/10.46714/735925LPI) and Simons Foundation 812811 to L.E. (https://doi.org/10.46714/735923LPI), and NSF Division of Biological Science SBS Biodiversity: Discovery and Analysis program (1753661) to B.J.B. This work made use of the Dutch national e-infrastructure with the support of the SURF Cooperative using grant no. EINF-2953. to T.E.

**Author contributions** T.J.G.E. conceived and supervised the study. A.S., K.U.K., W.H.L., Z.-S.H., A.-L.R., W.-J.L., T.N., M.B.S. and A.P.T. collected and provided environmental samples. E.F.C., F.H., J.H.S., N.D., K.W.S., B.J.B., L.-X.C., J.F.B. and E.S.J. performed metagenomic sequence assemblies and metagenomic binning analyses. L.E., D.T., E.F.C., C.W.S., K.W.S., J.L., B.J.B. and T.J.G.E. analysed the genomic data. L.E., D.T., E.F.C. and F.H. performed phylogenomic analyses. L.E., D.T., E.F.C., C.W.S., J.L. and T.J.G.E. investigated ESPs. E.F.C., L.E. and M.E.S. performed ancestral genome reconstruction analyses. V.D.A., C.W.S, B.J.B., L.E. and T.J.G.E. carried out metabolic inferences. L.E., D.T., E.F.C., C.W.S., V.D.A., B.J.B. and T.J.G.E. wrote, and all authors edited and approved, the manuscript.

**Funding** Open access funding provided by Uppsala University.

**Competing interests** The authors declare no competing interests.

**Additional information**
**Correspondence and requests for materials** should be addressed to Thijs J. G. Ettema.

| Family | Order | Class |
|---|---|---|

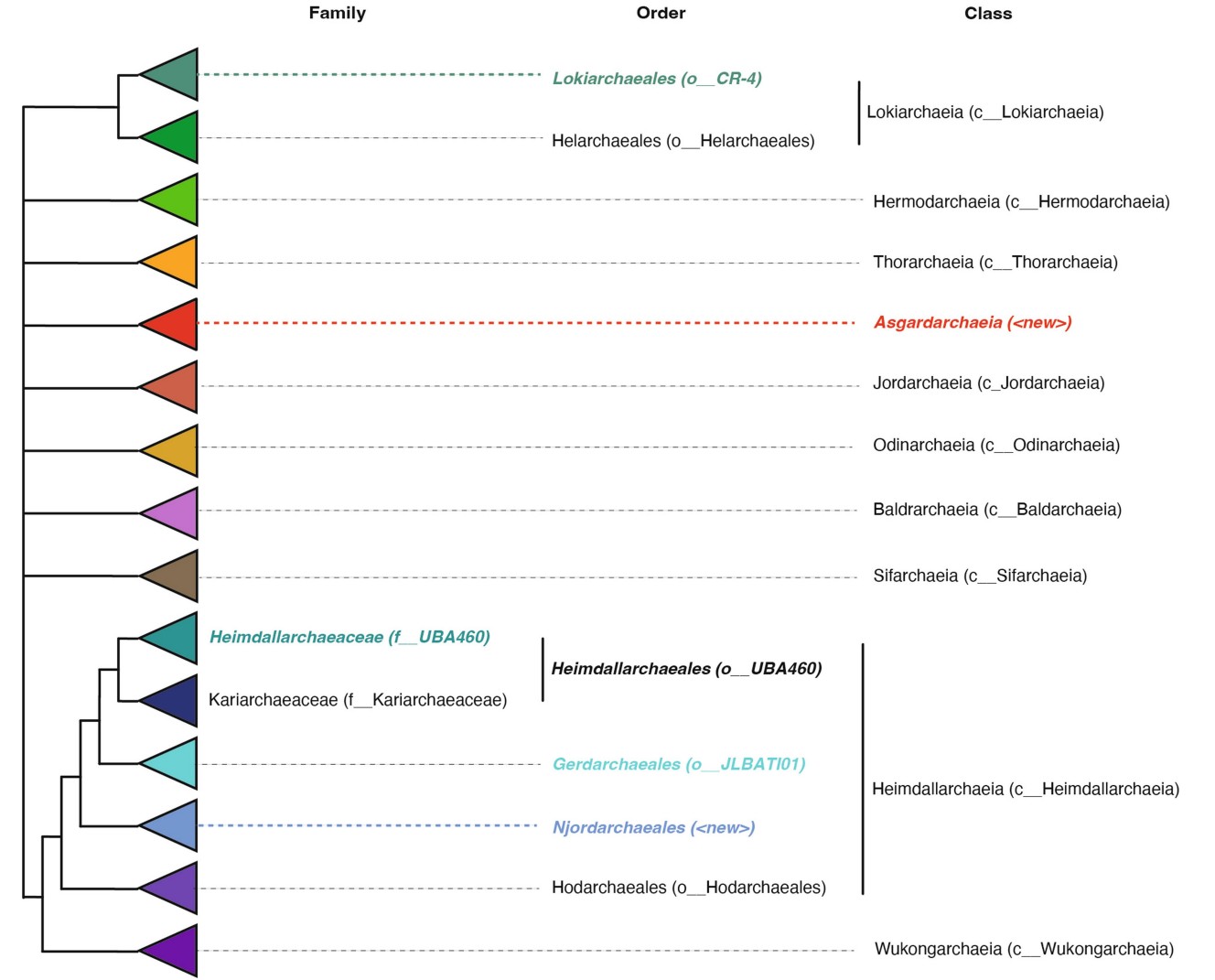

*Lokiarchaeales (o__CR-4)*

Helarchaeales (o__Helarchaeales)

Lokiarchaeia (c__Lokiarchaeia)

Hermodarchaeia (c__Hermodarchaeia)

Thorarchaeia (c__Thorarchaeia)

*Asgardarchaeia (<new>)*

Jordarchaeia (c_Jordarchaeia)

Odinarchaeia (c__Odinarchaeia)

Baldrarchaeia (c__Baldrarchaeia)

Sifarchaeia (c__Sifarchaeia)

*Heimdallarchaeaceae (f__UBA460)*

Kariarchaeaceae (f__Kariarchaeaceae)

*Heimdallarchaeales (o__UBA460)*

*Gerdarchaeales (o__JLBATI01)*

*Njordarchaeales (<new>)*

Hodarchaeales (o__Hodarchaeales)

Heimdallarchaeia (c__Heimdallarchaeia)

Wukongarchaeia (c__Wukongarchaeia)

**Extended Data Fig. 1 | Cladogram of proposed taxonomic scheme for the ranks of family, order and class for Asgard archaeal lineages employed in this study.** Equivalent names in GTDB are shown in parentheses. Cases with differing or new names have been highlighted in colored bold italics.

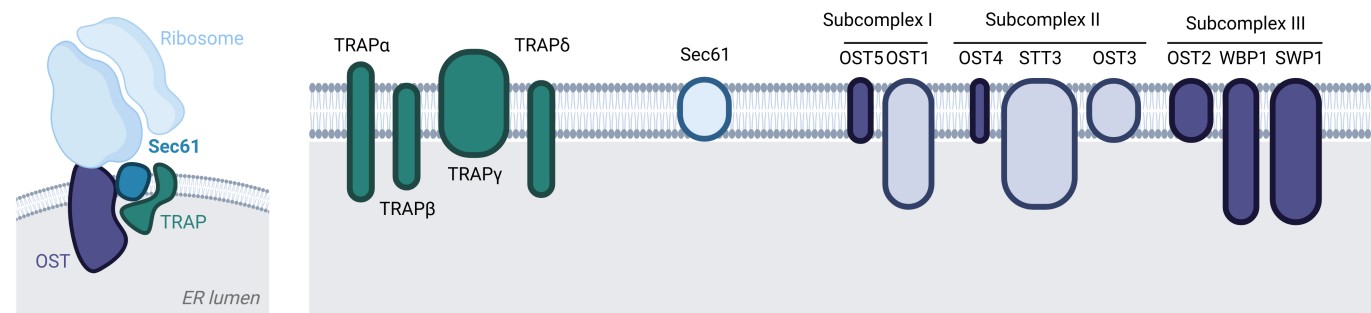

**Extended Data Fig. 2 | Asgard archaea encode homologs of eukaryotic protein complexes involved in N-glycosylation.** The Sec61, the OST and TRAP complexes are depicted according to their eukaryotic composition and localization. On the right-hand side of the panel, dark-colored subunits represent eukaryotic proteins which have prokaryotic homologs in Asgard archaea newly identified as part of this work; Light-colored subunit homologs have been described previously[3]. Figure generated using BioRender (https://www.biorender.com).

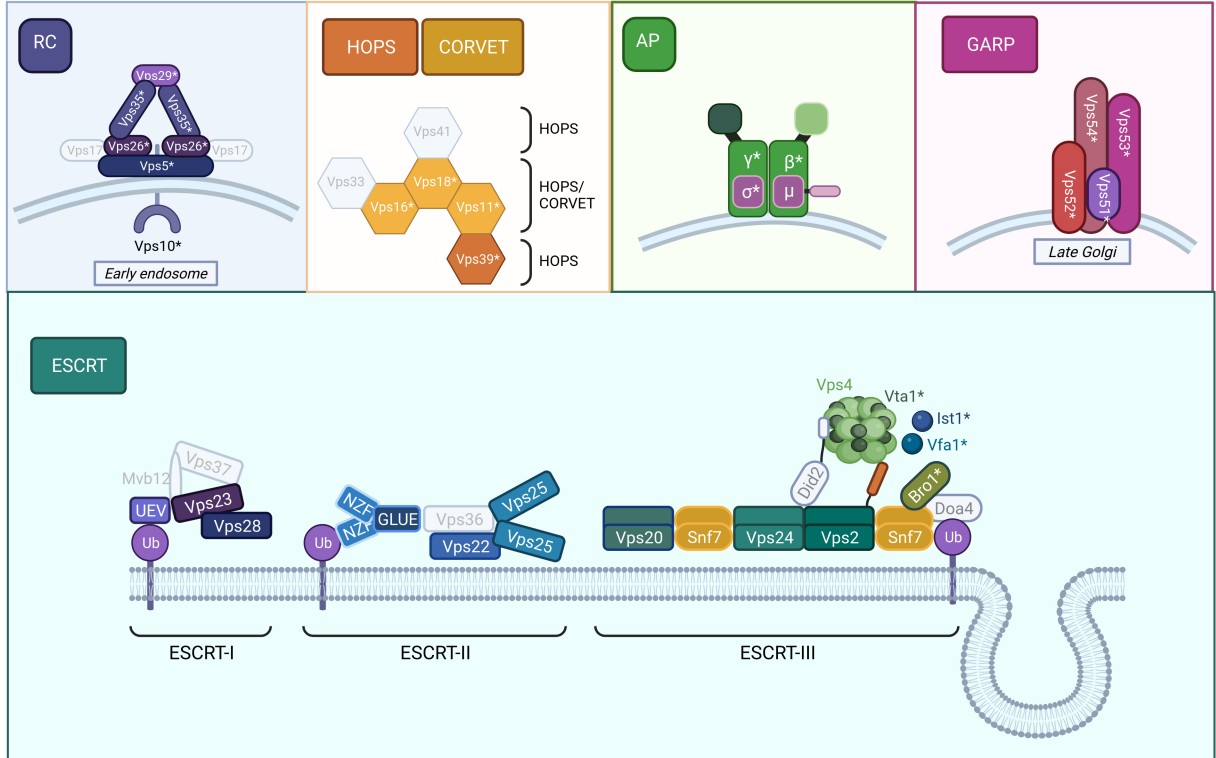

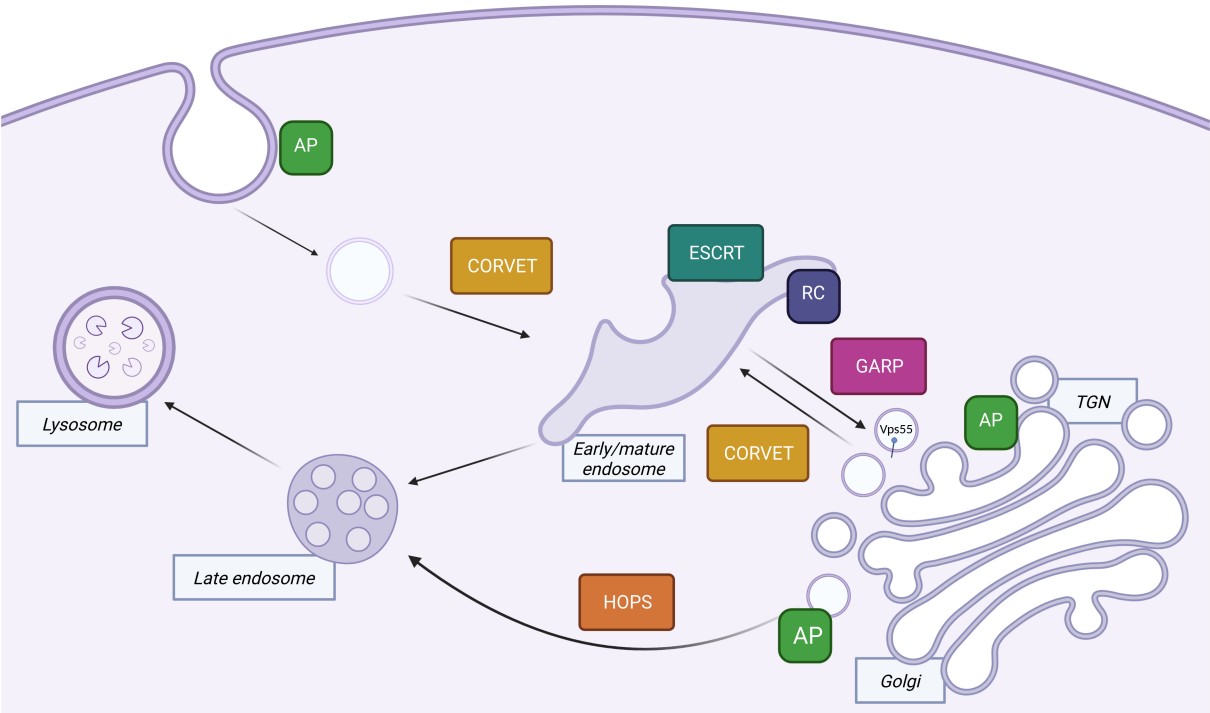

**Extended Data Fig. 3** | See next page for caption.

**Extended Data Fig. 3 | Identification of previously undetected vesicular trafficking ESPs in Asgard archaea.** Schematic representation of a eukaryotic cell in which ESPs involved in membrane trafficking and endosomal sorting that have been identified in Asgard archaea are highlighted. Colored subunits have been detected in some Asgard archaea while grey ones seem to be absent from all current representatives. Only major protein complexes are depicted. Additional components can be found in Fig. 2. From left to right, top to bottom: RC, Retromer complex. Retromer is a coat-like complex associated with endosome-to-Golgi retrograde traffic[35]. It is formed by Vacuolar protein sorting-associated protein 35, Vps5, Vps17, Vps26 and Vps29[118]. During cargo recycling, retromer is recruited to the endosomal membrane via the Vps5-Vps17 dimer. Cargo recognition is thought to be mediated primarily through Vps26 and possibly by Vps35. Finally, the BAR domains of Vps5-Vps17 deform the endosomal membrane to form cargo-containing recycling vesicles. Their distribution is sparse, but we have detected Asgard archaeal homologs of all subunits except for Vps17. Interestingly, the Thorarchaeia Vps5-BAR domain is often fused to Vps28, a subunit of the ESCRT machinery complex I, suggesting a functional link between BAR domain proteins and the thorarchaeial ESCRT complex. The best-characterized retromer cargo is Vps10. This transmembrane protein receptor is known in yeast and mammal cells to be involved in the sorting and transport of lipoproteins between the Golgi and the endosome. The Vps10 receptor releases its cargo to the endosome and is recycled back to the Golgi via the retromer complex[119]. CORVET: Class C core vacuole/endosome tethering complex; HOPS: Homotypic fusion and protein sorting complex. Endosomal fusion and autophagy depend on the CORVET and HOPS hexameric complexes[37]; they share the core subunits Vps11, Vps16, Vps18, and Vps33[120]. In addition, HOPS is composed of Vps41 and Vps39[121]. Vps39, found associated to late endosomes and lysosomes, promotes endosomes/lysosomes clustering and their fusion with autophagosomes[122]. AP, Adaptor Proteins. Asgard archaea genomes from diverse phyla encode key functional domains of the AP complexes. The eukaryotic AP tetraheteromeric structure is depicted, each color corresponding to a PFAM functional domain (Medium green: Adaptin, N terminal region; Dark green: Alpha adaptin, C-terminal domain; Light green: Beta2-adaptin appendage, C-terminal sub-domain; Dark pink/clear outline: Clathrin adaptor complex small chain; Light pink/dark outline: C-ter domain of the mu subunit); all five domains were detected in Asgard archaea, although not fused to each other. GARP: Golgi-associated retrograde protein complex. The GARP complex is a multisubunit tethering complex located at the trans-Golgi network where it functions to tether retrograde transport vesicles derived from endosomes[36,123]. GARP comprises four subunits, VPS51, VPS52, VPS53, and VPS54. ESCRT: Endosomal Sorting Complex Required for Transport system. This complex machinery performs a topologically unique membrane bending and scission reaction away from the cytoplasm. While numerous components of the ESCRT-I, II and III systems have been previously detected in Asgard archaea[2,3,38], we here report Asgard homologs for several ESCRT-III regulators Vfa1, Vta1, Ist1, and Bro1. The bottom panel shows where these complexes mainly act in eukaryotic cells. Ub: Ubiquitin; Vps: vacuolar protein sorting. Subunit names in grey indicate that no homologs were detected in Asgard archaea. Domains newly identified as part of this study are indicated with an asterisk. Figure created using BioRender (https://www.biorender.com).

# nature research

# Reporting Summary

Nature Research wishes to improve the reproducibility of the work that we publish. This form provides structure for consistency and transparency in reporting. For further information on Nature Research policies, see our Editorial Policies and the Editorial Policy Checklist.

## Statistics

For all statistical analyses, confirm that the following items are present in the figure legend, table legend, main text, or Methods section.

| n/a | Confirmed | |
|---|---|---|
| ☐ | ☒ | The exact sample size (*n*) for each experimental group/condition, given as a discrete number and unit of measurement |
| ☒ | ☐ | A statement on whether measurements were taken from distinct samples or whether the same sample was measured repeatedly |
| ☐ | ☒ | The statistical test(s) used AND whether they are one- or two-sided<br>*Only common tests should be described solely by name; describe more complex techniques in the Methods section.* |
| ☒ | ☐ | A description of all covariates tested |
| ☒ | ☐ | A description of any assumptions or corrections, such as tests of normality and adjustment for multiple comparisons |
| ☐ | ☒ | A full description of the statistical parameters including central tendency (e.g. means) or other basic estimates (e.g. regression coefficient) AND variation (e.g. standard deviation) or associated estimates of uncertainty (e.g. confidence intervals) |
| ☐ | ☒ | For null hypothesis testing, the test statistic (e.g. *F*, *t*, *r*) with confidence intervals, effect sizes, degrees of freedom and *P* value noted<br>*Give P values as exact values whenever suitable.* |
| ☐ | ☒ | For Bayesian analysis, information on the choice of priors and Markov chain Monte Carlo settings |
| ☒ | ☐ | For hierarchical and complex designs, identification of the appropriate level for tests and full reporting of outcomes |
| ☒ | ☐ | Estimates of effect sizes (e.g. Cohen's *d*, Pearson's *r*), indicating how they were calculated |

*Our web collection on statistics for biologists contains articles on many of the points above.*

## Software and code

Policy information about availability of computer code

Data collection   No software was used for data collection.

Data analysis   Custom scripts have been deposited on Github (https://github.com/laurajjeme/phylogenetics). Published software used for data analysis include BBTools v38.79,  Sickle v1.33, metaSPAdes v3.10.1, MetaBAT v2.12.1, Trimmomatic v.0.36, MEGAHIT v.1.1.1-2-g02102e1, SeqTK v1.0r75, CONCOCT v0.4.1, CLARK v1.2.3, miComplete v1, mmgenome v0.7.1, IDBA-UD 1.1.3, cutadapt v1.12, CheckM v1.0.5, IQ-TREE v. 2.0-rc2, Prokka v1.12, SiLiX v.1.2.10, Hifix v1.0.6, HHblits v3.0.3, Interproscan 5.25-64.0, EggNOG mapper v0.12.7, GhostKoala,

For manuscripts utilizing custom algorithms or software that are central to the research but not yet described in published literature, software must be made available to editors and reviewers. We strongly encourage code deposition in a community repository (e.g. GitHub). See the Nature Research guidelines for submitting code & software for further information.

## Data

Policy information about availability of data

All manuscripts must include a data availability statement. This statement should provide the following information, where applicable:

- Accession codes, unique identifiers, or web links for publicly available datasets
- A list of figures that have associated raw data
- A description of any restrictions on data availability

The MAGs reported in this study have been deposited at DDBJ/EMBL/GenBank. BioProject IDs, BioSample IDs and GenBank assembly accession numbers are available in Supplementary Table 1. All raw data underlying phylogenomic analyses (raw and processed alignments and corresponding phylogenetic trees), and all predicted proteomes have been deposited on Figshare (10.6084/m9.figshare.22678789).

# Field-specific reporting

Please select the one below that is the best fit for your research. If you are not sure, read the appropriate sections before making your selection.

☒ Life sciences ☐ Behavioural & social sciences ☐ Ecological, evolutionary & environmental sciences

For a reference copy of the document with all sections, see nature.com/documents/nr-reporting-summary-flat.pdf

# Life sciences study design

All studies must disclose on these points even when the disclosure is negative.

| | |
|---|---|
| Sample size | Sample sizes (i.e. the number of lineages included in phylogenomic analyses) were empirically determined based on the computational resources necessary to run the various analyses. |
| Data exclusions | No data was excluded |
| Replication | Robustness and reliability of phylogenetic analyses were assessed using 100 bootstrap replicates for all maximum likelihood analyses, as is commonly done in the field. |
| Randomization | Randomization is not necessary to a study using phylogenetic approaches because these approaches rely on the comparison of evolutionary relationships between species, rather than on random assignment of treatments or control groups. Phylogenetic analyses are not affected by the same sources of bias as experimental designs, such as confounding variables or selection bias. Therefore, while randomization is a useful tool in many types of research, it is not essential in studies that using phylogenetics and comparative genomics. |
| Blinding | Blinding is not necessary to a study using phylogenetic approaches because these methods are based on objective comparisons of evolutionary relationships between species, rather than on subjective assessments or measurements of treatment effects. These analyses do not involve human subjects, interventions, or subjective judgments that could be influenced by knowledge of the study conditions or treatments. Therefore, blinding is not relevant to the validity or reliability of phylogenetic studies, and its use is not required or expected in this type of research. |

# Reporting for specific materials, systems and methods

We require information from authors about some types of materials, experimental systems and methods used in many studies. Here, indicate whether each material, system or method listed is relevant to your study. If you are not sure if a list item applies to your research, read the appropriate section before selecting a response.

## Materials & experimental systems

| n/a | Involved in the study |
|---|---|
| ☒ | ☐ Antibodies |
| ☒ | ☐ Eukaryotic cell lines |
| ☒ | ☐ Palaeontology and archaeology |
| ☒ | ☐ Animals and other organisms |
| ☒ | ☐ Human research participants |
| ☒ | ☐ Clinical data |
| ☒ | ☐ Dual use research of concern |

## Methods

| n/a | Involved in the study |
|---|---|
| ☒ | ☐ ChIP-seq |
| ☒ | ☐ Flow cytometry |
| ☒ | ☐ MRI-based neuroimaging |

