## [Peer Review File · Nature]

Manuscript Title: Inference and reconstruction of the heimdallarchaeial ancestry of eukaryotes

Reviewer Comments & Author Rebuttals

Reviewer Reports on the Initial Version:

Referee expertise:

Referee #1: eukaryogenesis

Referee #2: archaea

Referee #3: metagenomics

Referees' comments:

Referee #1 (Remarks to the Author):

Summary of the key results

This paper describes a large expansion in the number of known ASGARD archaea, clarifies some of the hitherto eukaryotic features that these genomes contain and propose the erection of new groupings on the basis of a variety of phylogenetic analyses.

Originality and significance: if not novel, please include reference

The biggest problem with this paper is a little unfortunate - the recent paper in Nature from Liu et al as also described many new ASGARD genomes and has also proposed the erection of new groupings and the identification of features in these genomes that might previously have been thought to be eukaryotic-specific.

<https://www.nature.com/articles/s41586-021-03494-3>

I cannot support the independent publication of these two papers because of the overlap. Publishing this paper in its current format would only introduce confusion. In addition, when I compared the phylogenies of the current manuscript and the published paper from Liu et al, I struggle to see how the phylogenies are compatible. In fact, there seems to be some really major differences between these trees. Therefore - and I do sympathise with the current authors - I feel that until the other genomes are included in the analysis, then we should not proceed with publication of the current manuscript. I feel that it is not right to approve the introduction of confusion into the literature.

Data & methodology: validity of approach, quality of data, quality of presentation

I had a problem with some of the methodology. The authors state that "Careful inspection of the obtained RP56 tree however revealed a potential artefact: Idunnarchaeota, considered bona fide Asgard archaea based on the presence of many encoded 'typical' Asgard-like ESPs¹⁰, were found to branch outside of the Asgard archaea, at the base of the TACK superphylum and as a sister to Korarchaeota." The authors conclude that this placement 'might' be due to compositional problems with ribosomal proteins. They then reject the entire dataset and decide to use a different dataset. I'm not particularly happy with this line of thinking. It feels like it is circular reasoning - "we didn't like the topology, so we decided the dataset was the problem, so we used a different dataset, which gave us a topology that we were happier with, so this dataset is better". I would need some better reasoning than this.

The data quality seems fine and the quality of the presentation is fine.

Appropriate use of statistics and treatment of uncertainties

As I have said in the previous passage, I have a problem with the statistical approach, which seems to be based upon rejecting datasets that don't give the acceptable answer and instead using a dataset that does give the acceptable answer.

Conclusions: robustness, validity, reliability

Until the Liu et al data is included, then I don't believe we have a clear current view of the ASGARD Archaea and the Eukaryotes.

Suggested improvements: experiments, data for possible revision

As I have said - inclusion of all known ASGARD sequences and a clear and statistically sound reason for using one dataset over another in order to construct phylogenies is needed.

References: appropriate credit to previous work?

Yes, this is fine.

Clarity and context: lucidity of abstract/summary, appropriateness of abstract, introduction and conclusions

Yes, the writing is clear and easy to read.

Referee #2 (Remarks to the Author):

The manuscript by Eme et al. reports the reconstruction and analysis of 63 new metagenome-assembled-genomes (MAGs) belonging to the Asgard clade, the closest archaeal relatives of eukaryotes. These data significantly expand previous views on the diversity of this superphylum and identifies four additional candidate phyla. The authors proceed to analyze these MAGs plus 17 already available ones.

The first major result is the robust affiliation of eukaryotes within the Heimdallarchaeota, and more specifically to class Skadiarchaeia. This a very important finding, revealing that the more we explore the diversity of the Asgard, the closer we get to the first eukaryotic cell, providing key elements to understand the events that led to eukaryogenesis. Strong of this larger sampling, the authors then move on to reconstruct the gene content of the last common ancestor of all Asgards and study the dynamics of gene loss and duplications during their diversification. Interestingly, this reveals significantly more events of duplications than losses, akin what happened in early eukaryotic evolution, providing evidence for an additional shared characteristic. Finally, inference on the lifestyle of the last common Asgard indicate it was a thermophilic chemolithotroph, and that subsequent adaptation to mesophilic environments and heterotrophy paved the way to eukaryogenesis.

I think this contribution is exceptional -both for the quantity and quality of the results, and certainly deserves publication. What I liked particularly is the phylogenetic analysis presented, which is very thorough, uses the best practices in current phylogenomics (recoding, different models, and taxonomic sampling), and makes the results extremely trustworthy. Reconstructing deep evolutionary relationships is not a simple matter of "press button" but must be analyzed in the most accurate way, by testing all possible sources of error. Here, the authors convincingly show how ribosomal proteins contain a complex signal that likely produces tree-reconstruction artefacts (notably known to be linked to biases due to convergent adaptation to hyperthermophily for example) and identify an alternative dataset of 57 non-ribosomal markers that is much more robust. Therefore, this paper will represent in my opinion the last "nail in the coffin" in support of a 2D Tree of Life, it'll sweep away current debate mostly fueled by inappropriate analyses and will be referred to as such by the scientific community. This is a big advancement with respect to the recent Koonin and Li paper, which left the emergence of eukaryotes unresolved.

A few comments to improve readability and impact:

With the current turmoil about high-rank taxonomy, please explain in the main text the criteria you used to define the new phyla you're proposing. Does this hold when using the alternative 57 non-ribosomal markers?

Figure 2b, legend: it's in the sup mat but it would be nice to have here in parenthesis the number of positions left in the concatenation at each treatment.

Concerning the ESP, please clarify in the main text how you searched for them, it is a bit unclear (page 10).

Here, a comparison with the results of Koonin and Li would be necessary, I think.

Concerning your gene-tree/species-tree reconciliation and the reconstruction of the ancestral features of LAECA, I am not sure I understand why you included TACK and Euryarchaeota and not only Asgard? How do your results correlate to the ones by Williams et al on the root of archaea? I do not remember if they had Asgard in that analysis already. This is just a curiosity, and by no means essential to the paper. Line 968 in M&M, please say how you assembled the protein families.

Concerning the inferred higher duplication rates, I am sure many readers will ask if this is not due to incorrect binning and double representation of genes in the reconstructed MAGs. It would be good to mention and exclude this problem in the main text. For example, it could be checked rapidly on the families specifically duplicated in Skadi, if they are 100 identical or contaminations.

If this paper is accepted, please be careful to deposit all raw data, not only the MAGs analyzed here. This is very important.

Also, I think you'll need to coordinate with Koonin, Li, and Rinke on how to name these new phyla.

Referee #3 (Remarks to the Author):

A: Eme, Tamarit, Caceres et al. present a thorough and careful reconstruction of the Asgard archaeal superphylum, including a detailed examination of the branching of Eukaryotes within this group, based on sophisticated methodologies. The authors examine Eukaryotic signature proteins, identifying several new types represented within Asgardian genomes, and reconstruct predictions for ancestral life styles for both LAsCA and LAECA. The manuscript is clearly written and the information presented represents a significant advance in our understanding of the diversity within the Asgard superphylum and the nature of the relationship of the Asgard archaea with the Eukaryotes, from an evolutionary perspective.

B: Originality and significance: The work complements recent work examining this radiation and its relationship to Eukaryogenesis, with a more detailed reconstruction of ancestral phenotypes and different ESPs identified. It is a valuable and impressive contribution to a rapidly maturing area of research.

C: Data and methodology: The data is handled expertly, and the quality of the data is high. There are several points where the authors draw conclusions that are not fully supported by the data, or which appear selected based on a favoured scenario, which need to be tempered or more strongly justified before the paper would be acceptable for publication.

D: Appropriate use of statistics and treatment of uncertainties: Appropriate tests to examine divergences or overlaps (e.g., amino acid frequency data) are used to bolster conclusions. Uncertainty is occasionally under-reported in the main text compared to the supplement, leading to overly strong conclusions.

E: Conclusions: as stated, the general conclusions and new data are robust and exciting. The

conclusions around more specific relationships or metabolic reconstructions are in some cases too speculative.

F: Suggested improvements: see below for my detailed comments and suggested edits.

G: Referencing is appropriate and complete.

H: Clarity and context: The paper is written clearly and concisely, with substantial supplemental data to bolster the presented work.

My detailed comments follow below:

Major comments:

1) Data is not available. Supplemental table 1 lists the 64 new MAGs, but only 9 have MAG-related accessions, and none of the Biosample accessions for the other MAGs that I checked were accessible on NCBI. These should have been available at the time of submission.

2) Originality and novelty: There have been several recent papers/pre-prints expanding the Asgard radiation, and naming new clades. Recent publications expanding the Asgard lineage and naming phyla require reassessment of the proposed names in this work – are any of the Gefion-, Freya-, Vidar-, and Idunnarchaeota the same lineages as described in Liu et al., Nature 2021 (<https://www.nature.com/articles/s41586-021-03494-3>) and if so, please adjust nomenclature accordingly.

I appreciate that these genomes were not available at the time of submission, but it would be irresponsible not to assess overlaps in novel clades prior to the formal namings presented in the supplemental materials being accepted.

3) How are phyla being determined? From phylogenies, 11 lineages of Asgard does not appear commensurate with phylum-level lineages in other archaeal radiations, by sequence distance. Idunn is clearly a new group, but the others are less clear aside from inconsistent associations with other lineages – the rationale for phylum determination is not presented.

4) The instability of the Asgard lineages with respect to each other suggests resolving power is limited and the deeper divergence to the Euks also unstable – it is difficult to understand why a single answer (sister to Skadiarchaea) was selected as correct given the robust trees under alternate hypotheses are equally well supported/poorly supported, and the phyloBayes did not converge – that alone suggests there is not sufficient information to resolve the specific branch point. This is a major point of the paper, and I think the conclusion (the title of the paper) is over stepping the data. Similarly: Supplemental materials Page 27, line 585 – did the Phylobayes for NM57 datasets converge? If not, what were the topologies observed along the different chains? The treatment of the datasets and phylogenetic signals seems unequal – lack of convergence is listed as a reason to doubt the RP56 data, but not the NM57. Euks are unresolved across RP56 treatments, but equally unresolved in NM57 cases, where one treatment is selected as the correct inference. Stronger justification for selecting one option as correct is needed throughout. The justification of

Korarchaeota/Idunnarchaeota attraction based on amino acid composition is sufficient for those decisions, but does not speak to the placement of Euks in trees experiencing that artifact.

5) the CIII complex in Skadiarchaea – activity in conjunction with a Rieske center is highly speculative, and based on one protein undertaking a role it is not normally associated with (NarI passing electrons to Rieske iron-sulfur protein) and a poorly resolved prediction – “potential Rieske iron sulfur protein”, which is also suggested to play a role in electron transfer from NarI to cupredoxin. Cupredoxin itself has not been seen as an electron shuttle in archaea, adding another layer of conjecture to this reconstruction. This, of all hypothesized metabolic and energetic functions, is the least convincing, and is featured in main text figure 4. I feel this is too speculative to be given such weight in the main document.

6) Throughout, there are different datasets in use, and the descriptions of these datasets are not clearly presented. I recognize this is a collaborative work, likely with different aspects occurring in parallel, but which dataset is which, and which of the novel MAGs were included in each is not well telegraphed in the text.

Examples:

Page 6 line 64 – why only 17 additional Asgard MAGs? There are more available, and for deep tree reconstructions, shorter branches are advantageous. What constituted “phylogenetically diverse”? If these were selected specifically for 16S rRNA genes associate with MAGs, that should be stated clearly. Your later dataset (page 7, line 90) contained 64 Asgard MAGs before the addition of the new 63 MAGs, so this disconnect between dataset completion is confusing.

Page 7 line 90 – updated by including additional MAGs, or some other form of updating? If addition, please clarify if the numbers of MAGs stated for the different groups are the current or original set. If current, why only 64 MAGs for Asgard, and were these pre-existing publicly available MAGs, or the newly described MAGs from this study, or a mix?

Page 39, line 736 – If all archaeal genomes were downloaded and one representative per species-level cluster was used, how are the three archaeal lineages acting as an outgroup? They are not outside the Archaea. Did you mean all Asgard archaeal genomes?

Please make explicit which genomes were added to this dataset from your new MAGs, as there is some confusion from the main text as to dataset sizes and overlaps.

Page 40, lines 749-754 – additional detail is needed here to clarify what the dataset underlying this tree is – the addition of the NCBI Asgard to the previous dataset? Is this an entirely new dataset, with a different reference set or outgroup?

Page 42, line 786 – which dataset is this – the original tree, the NCBI-Asgard-included tree, or another? How many of those Asgard MAGs were from this study? I suggest developing names for your different datasets, and making explicit what is contained in each one.

7) Page 48, line 918 – what is the justification for: “These protein pairs were joined by concatenating the amino acid sequence of the proteins containing beta-propeller repeats (N-terminus) and alpha-solenoid repeats (C-terminus)” for proteins that are up to 10 genes apart from each other on the genome? Is there evidence of such distant interactions for other proteins within these (or other) genomes? Are the genes operonic with each other? This is a stretch compared to the other protein families examined, especially given “individual β -propeller and α -solenoid folds are commonly found among prokaryotes” (supp material page 21, line 453). Are there similar pairs in other, non-Asgard,

genomes that are co-located and receive similar support under a concatenated protein structure analysis?

Supplemental figure 25 – the gene neighbourhoods do not dispel my concerns around positing protein fusions/complexing for these genes, which are largely non-operonic.

Minor comments

Main text:

Page 13 line 228 – give the median for % genome for small GTPases in place of, or at least, alongside the maximum.

Page 13 line 238 – missing the word “of”: of the gamma subunit.

Page 15 – is the increase in proteome size for the Skadia solely due to gene duplications? What role is LGT afforded in your models?

Page 16 line 307 – your data supports this prediction, it does not confirm it.

Page 19 line 365 – sentence starting “Of these, Asgard archaeal proteins...” is difficult to follow. Please revise.

Page 20 ,line 381 – consider if “maintained” is the correct word here?

Figure 1 line 514 – using “at least 5 proteins” out of 15 targeted is below the recommended sequence completion for a concatenated gene alignment – ideally taxa have 50% presence within alignment sites, otherwise long branch effects are artefactually observed.

Figure 2 – what was the rationale for selected Asgard MAG inclusion here? Only a small fraction of the expanded dataset is used, and while I appreciate the rigor of the phylogenetic analyses, and the presentation of alternate, supported relationships, the taxon selection needs better justification.

Figure 4 – “The Wood-Ljungdahl pathway (WLP) appeared only to be present in the LAsCA and was lost in the more recent ancestors examined here.” – consider rephrasing, as you examine all Asgard lineages so “here” could mean the manuscript, or only Heimdall and Idunn in this figure. As written it implies WLP is missing from all extant Asgard lineages, and that makes reconstruction of the ancestor overly speculative.

Methods

Page 41 lines 780-783 – how were annotations from different servers integrated?

Page 43, line 816 – sentence/phrase starting “we kept only the ones..” is grammatically incorrect, I think some words are out of order.

Fast site removal – trialed 10-90% removal, which was used for the final tree? I'm not sure that was stated explicitly.

Page 48 – Ancestral reconstruction – as before, please make explicit which genomes were included, what role the newly described MAGs play in this analysis, and how genomes were selected for inclusion.

Page 49 line 939 - is the threshold of 0.3 a proportion or a different statistic? Inferring presence ancestrally with 30% presence within a family seems low. Is there precedence for this threshold? Also, the sentence says 0.3 “for an event to be considered” – how were decisions ultimately made? What were the empirical requirements? On line 943 it also states “considered if its copy number was above 0.3” – considered, or accepted? Considered implies a later decision process that is not described.

Supplemental materials

Page 22, line 480 – font change, this section also lacks references where appropriate.

Page 24, line 517 – similarly, references or data (figure mention) lacking for Yip section

Page 25 line 554 – what makes a BAR domain a “clear BAR domain” – threshold of similarity? Scores? Please clarify.

Page 33, line 736 – sentence “As expected, several ‘patchy’ ESPs currently only found in few Asgard archaeal genomes, such as tubulin (only present in Odinararchaeota genomes), are predicted to be absent in LAECA-proxy nodes, such as Sec23/Sec24, TRAPP and ubiquitin-like proteins.” Has two sets of examples, interspersing the main sentence – is confusing, suggest rephrasing. Are Sec23/Sec24 etc. the LAECA-proxy nodes? I think they are not, but rather examples of patchy ESPs – not as written currently.

Page 34 line 757 – “Other homologous protein families” – how many? Given you provide 4 examples, and all 4 are then inferred to be present – how many were not considered likely present in LAECA?

Page 35 line 784 – “nearly most” is not informative, provide a proportion or precise number

Page 44, line 1000 – “A recent study...” – reference is missing.

Page 49 line 1142 – “could not identified detect” – remove identified.

Supplemental figure 9 is much less well formatted than the preceding trees.

Supplemental figure 11 is completely unreadable – the resolution prohibits any meaningful examination of the trees presented.

Supplemental figure 17 – what are the scales of the dots (content/originations)? Missing from legend.

Supplemental figure 23 has space for the bin names/scaffold names using location codes – the symbols are very small and difficult to distinguish even at maximum zoom, plus they do not add clarity to the figure.

Supplemental figure 32 is not described clearly enough to interpret. I have tried, and I genuinely do not know what numbers correspond to what comparisons/statistics, or what the trend line (?) on scatter plots are, among other confusions. Needs clearer axes, full figure legend.

Author Rebuttals to Initial Comments:

Referees' comments:

Referee #1 (R1):

R1.1: Summary of the key results

This paper describes a large expansion in the number of known ASGARD archaea, clarifies some of the hitherto eukaryotic features that these genomes contain and propose the erection of new groupings on the basis of a variety of phylogenetic analyses.

Originality and significance: if not novel, please include reference

The biggest problem with this paper is a little unfortunate - the recent paper in Nature from Liu et al as also described many new ASGARD genomes and has also proposed the erection of new groupings and the identification of features in these genomes that might previously have been thought to be eukaryotic-specific.

<https://www.nature.com/articles/s41586-021-03494-3>

I cannot support the independent publication of these two papers because of the overlap. Publishing this paper in its current format would only introduce confusion. In addition, when I compared the phylogenies of the current manuscript and the published paper from Liu et al, I struggle to see how the phylogenies are compatible. In fact, there seems to be some really major differences between these trees. Therefore - and I do sympathise with the current authors - I feel that until the other genomes are included in the analysis, then we should not proceed with publication of the current manuscript. I feel that it is not right to approve the introduction of confusion into the literature.

Response: We would like to thank the reviewer for evaluating our work. We perfectly understand their perspective, and we have now included in our analyses all the genomes published as part of the Liu et al. paper, which supports and reinforces our previous results regarding the placement of eukaryotes.

We have also made an effort to harmonize clade names in two ways: 1) using the nomenclature proposed in Liu et al. when there was overlap and highlighting the novel clades introduced here; 2) using a standardized approach to define taxonomic ranks, as proposed by Rinke et al (2021, Nat Microbiol 6: 946) and more recently applied to Asgard archaeal taxonomy in Sun et al. (2021, ISME Comm 1: 30) and the last versions of the Genome Taxonomy Database (GTDB). We include both of these two systems in the supplementary material (Table S1, "MAGTaxonomy", columns "Liu et al (2021) taxonomy" and "Sun et al. (2021) taxonomy"), and expand on them by including new taxon roots and applying the taxonomic rank corresponding to each group's position

in the Asgard tree (red font in Table S2). To accommodate the current trends in archaeal taxonomic nomenclature, in this manuscript we have now used the second nomenclature system, based on the GTDB system. For example, the formerly proposed class Skadiarchaeia has now been renamed as the order “Hodarchaeales” due to 1) the genomic overlap with Liu et al’s phylum Hodarchaeota (i.e. containing the genome of *Hodarchaeum mangrovi* strain FT_5_011, genome type of Hodarchaeota in Liu et al (2021), and other genomes classified under this taxon), and 2) their placement as a subclade within the class Heimdallarchaeia in Rinke et al (2021) and Sun et al (2021). Additionally, as mentioned above, we are in the process of organizing a community-driven effort to establish a uniform/standardized taxonomic classification for Asgard archaea, with the aim to have this integrated in taxonomic reference databases such as GTDB, NCBI and Silva.

Finally, we discuss in supplementary material the methodological reasons for the discrepancies between the Liu et al. topologies and ours. First of all, it is important to note the phylogenetic analyses represented a limited (and by no means exhaustive) part of the paper by Liu and colleagues. It was not their main aim to carry out sophisticated analyses, and they acknowledged themselves that “Further phylogenomic study with an even broader representation of diverse archaeal lineages, extended sets of phylogenetic markers and—possibly—more sophisticated evolutionary models are required to clarify the relationships between archaea and eukaryotes.” Indeed, they only analyzed a dataset of 29 proteins with a fairly simple evolutionary model (LG+R10), a non-mixture model more prone to long-branch attraction artefacts. This is particularly problematic in the case of their analyses since they always included a bacterial outgroup, representing an extremely long branch in the tree susceptible to artificially attracting the long eukaryotic branch. This can explain why they recover eukaryotes at the base of the Wukong+Heimdall clade instead of placing them in a more nested position such as is recovered in our analyses.

R1.2: Data & methodology: validity of approach, quality of data, quality of presentation

I had a problem with some of the methodology. The authors state that "Careful inspection of the obtained RP56 tree however revealed a potential artefact: Idunnarchaeota, considered bona fide Asgard archaea based on the presence of many encoded 'typical' Asgard-like ESPs¹⁰, were found to branch outside of the Asgard archaea, at the base of the TACK superphylum and as a sister to Korarchaeota." The authors conclude that this placement 'might' be due to compositional problems with ribosomal proteins. They then reject the entire dataset and decide to use a different dataset. I'm not particularly happy with this line of thinking. It feels like it is circular reasoning - "we didn't like the topology, so we decided the dataset was the problem, so we used a different dataset, which gave us a topology that we were happier with, so this dataset is better". I would need some better reasoning than this.

Response: Even though we understand this point, we respectfully disagree with the reviewer that we used “circular reasoning”, or that we favor a particular topology. We would like to re-emphasize that several independent lines of evidence point to the RP56 dataset as problematic, particularly regarding the phylogenetic placement of Njordarchaeales (former Idunnarchaeota), which stems (among others) from compositional bias. Below, we

list four points supporting the idea that the RP56 dataset is not suited to retrieve the correct phylogenetic position of Njordarchaeales.

First of all, removing Korarchaeota from this RP56 dataset (and the RP15 dataset used in Fig. 1) results in recovering Njordarchaeales as sister lineage to (or within) Heimdallarchaeia (e.g. Figure S7), similar to what is observed for the NM57 dataset (with and without Korarchaeota (e.g. see Figure S17, S18 and S21), suggesting this dataset is less prone to carry sites attracting Njordarchaeales to Korarchaeota).

Second, we provide evidence in Figures S4-6 that the compositional bias is stronger in the RP56 dataset than in the NM57 dataset. In particular, we identified the individual sites in our alignments that favor one or the other topology (monophyly of Njordarchaeales + Korarchaeota, versus monophyly of Njordarchaeales + Heimdallarchaeia). We show that the amino acid composition of the former is strongly driven by adaptation to thermophily and that they present a statistically significantly stronger “thermophilic signature” than the sites supporting the monophyly of Njordarchaeales and Heimdallarchaeia (Supplementary Discussion 1.1.3.3.). Although we acknowledge that this does not mean that the topology change is single-handedly the result of this compositional bias, it is well-established that standard evolutionary models such as the LG+C60, which are homogeneous across branches, are not designed to accommodate such biases in particular parts of the tree, and that the stronger the bias, the more likely it will yield a reconstruction artefact.

Third, we observe that the individual Phylobayes chains ran on the RP56 dataset do carry this conflictual signal, with some of them showing Njord+Kor while some show Njord+Heimdall. This strongly suggests that this dataset does not reveal a different evolutionary history but instead generates a complex tree space with local optima, resulting from an important non-phylogenetic signal mixed in with the historical signal. In contrast, the chains run on the NM57 dataset are all consistent, supporting the monophyly of Njordarchaeales and other Heimdallarchaeia.

Finally, the distribution pattern of a number of ESPs supports the sister relationship of Njordarchaeales and other Heimdallarchaeia (e.g. ribosomal protein L28e) and are uniquely found in these lineages. The alternative interpretation of these results requires invoking a scenario involving horizontal gene transfer of these key informational proteins and simultaneously a complex, unexplained evolutionary history of the gene markers used as part of the RP56 and NM57 datasets that would artefactually place Njordarchaeales also within Asgard archaea. Thus, phylogenomics and gene content analyses are, together, strong evidence for the monophyly of Heimdallarchaeia as an Asgard archaeal group that includes Njordarchaeales.

Because of space constraints of the manuscript, many of these analyses are only presented in the supplementary material, but we have now more clearly outlined these issues pointed out above in the main text.

R1.2: The data quality seems fine and the quality of the presentation is fine.

Appropriate use of statistics and treatment of uncertainties

As I have said in the previous passage, I have a problem with the statistical approach, which seems to be based upon rejecting datasets that don't give the acceptable answer and instead using a dataset that does give the acceptable answer.

Response: We believe that, given the arguments developed above, the reviewer will hopefully agree with us that the arguments for favoring one dataset over the other are evidence-based and not due to subjective preferences.

R1.3: Conclusions: robustness, validity, reliability

Until the Liu et al data is included, then I don't believe we have a clear current view of the ASgard Archaea and the Eukaryotes.

Suggested improvements: experiments, data for possible revision

As I have said - inclusion of all known ASgard sequences and a clear and statistically sound reason for using one dataset over another in order to construct phylogenies is needed.

Response: Please see our response to the editor above. We have now included all Asgard sequences available as of May 12th, 2021, including those of the Lui et al. study, which confirm and further support our previous results. We have clarified and expanded on the reasons for favoring the NM topology over the RP one (see above). We nonetheless discuss in more detail the conflicting phylogenetic signal and make clear that several alternative hypotheses cannot be rejected. The main text now reads as:

“In summary, resolving the position of eukaryotes relative to Asgard archaea is anything but trivial (see Supplementary Discussion). In our efforts to extract the true phylogenetic signal, we provide confident support for eukaryotes forming a well-nested clade within the Asgard archaea phylum, consistent with the 2D tree of life scenario. More specifically, we observe that eukaryotes affiliate with the Heimdallarchaeia in analyses in which we systematically reduce phylogenetic artefacts, predominantly converging on a position of eukaryotes as sister to Hodarchaeales, which is also in line with the observed ESP content and genome evolution dynamics (see below).”

References: appropriate credit to previous work?

Yes, this is fine.

Clarity and context: lucidity of abstract/summary, appropriateness of abstract, introduction and conclusions

Yes, the writing is clear and easy to read.

Response: We would like to thank this reviewer again for the constructive feedback and for helping us improve our manuscript.

Referee #2 (R2):

R2.1: The manuscript by Eme et al. reports the reconstruction and analysis of 63 new metagenome-assembled-genomes (MAGs) belonging to the Asgard clade, the closest archaeal relatives of eukaryotes. These data significantly expand previous views on the diversity of this superphylum and identifies four additional candidate phyla. The authors proceed to analyze these MAGs plus 17 already available ones.

The first major result is the robust affiliation of eukaryotes within the Heimdallarchaeota, and more specifically to class Skadiarchaeia. This a very important finding, revealing that the more we explore the diversity of the Asgard, the closer we get to the first eukaryotic cell, providing key elements to understand the events that led to eukaryogenesis. Strong of this larger sampling, the authors then move on to reconstruct the gene content of the last common ancestor of all Asgards and study the dynamics of gene loss and duplications during their diversification. Interestingly, this reveals significantly more events of duplications than losses, akin what happened in early eukaryotic evolution, providing evidence for an additional shared characteristic. Finally, inference on the lifestyle of the last common Asgard indicate it was a thermophilic chemolithotroph, and that subsequent adaptation to mesophilic environments and heterotrophy paved the way to eukaryogenesis.

I think this contribution is exceptional -both for the quantity and quality of the results, and certainly deserves publication. What I liked particularly is the phylogenetic analysis presented, which is very thorough, uses the best practices in current phylogenomics (recoding, different models, and taxonomic sampling), and makes the results extremely trustworthy. Reconstructing deep evolutionary relationships is not a simple matter of "press button" but must be analyzed in the most accurate way, by testing all possible sources of error. Here, the authors convincingly show how ribosomal proteins contain a complex signal that likely produces tree-reconstruction artefacts (notably known to be linked to biases due to convergent adaptation to hyperthermophily for example) and identify an alternative dataset of 57 non-ribosomal markers that is much more robust. Therefore, this paper will represent in my opinion the last "nail in the coffin" in support of a 2D Tree of Life, it'll sweep away current debate mostly fueled by inappropriate analyses and will be referred to as such by the scientific community. This is a big advancement with respect to the recent Koonin and Li paper, which left the emergence of eukaryotes unresolved.

Response: We would like to thank the reviewer for their summary of our work and their overall positive feedback.

R2.2: A few comments to improve readability and impact:

With the current turmoil about high-rank taxonomy, please explain in the main text the criteria you used to define the new phyla you're proposing. Does this hold when using the alternative 57 non-ribosomal markers?

Response: We agree with the reviewer that the description of novel high-ranking taxa within Archaea has been somewhat arbitrary if not confusing, especially in regard to the recent plethora of archaeal MAGs. As indicated above (see our response to the editor), we have made an effort to justify and standardize the taxonomic naming scheme of Asgard archaea (as part of an ongoing community effort), which is based on GTDB criteria (see comment to editor above, Extended Data Figure 1 and Supplementary Information section 2.

“Unification of novel high-ranking taxon names with the Genome Taxonomy Database taxonomic scheme” for details).

In brief:

We included MAGs representing novel taxa from Liu et al (Nature, 2021) and Sun et al. (ISME Comm, 2021), and re-ran our phylogenies with both marker sets. We noted all previously named groups and confirmed their monophyly across datasets. We selected the GTDB naming scheme as of release 07-RS207 and updated our manuscript accordingly (see Supplementary Table S1). For the few cases in which we provided a name that replaces a temporary name in GTDB or refer to a group that has not yet been named in GTDB, we use the corresponding prefix accepted in the literature (i.e. Loki, Heimdall, Gerd, Kari and Njord) with the appropriate taxonomy suffix corresponding to their taxonomic rank in GTDB (i.e. -aceae, -ales, -ia) (Supplementary Table S1). In one case, we name a group that has not yet been published in previous literature, the Asgardarchaeia, but will be appropriately described in a community effort as part of an upcoming manuscript (Tamarit, Eme, Rinke, Baker, Ettema, et al., in prep). This manuscript includes new MAGs that help anchor the new group, novel phylogenomic efforts to place it with high confidence, and formal description for all Asgard archaeal taxa.

R2.3: Figure 2b, legend: it's in the sup mat but it would be nice to have here in parentheses the number of positions left in the concatenation at each treatment.

Response: Good point. We have now added the number of positions for each dataset, for each figure showing a phylogeny, in the main text and supplementary.

R2.4: Concerning the ESP, please clarify in the main text how you searched for them, it is a bit unclear (page 10).

Response: We have now clarified how ESPs were identified in more detail by adding the following into the methods section:

“In-depth analysis of potential ESPs involved a combination of automatic screens and manual curation. We first manually searched for homologues of previously described ESPs^{9,10,96} by using a variety of sequence similarity approaches such as BLAST, HMMer tools, profile-profile searches using HHblits, combined with phylogenetic inferences, and, in some cases, the Phyre2 structure homology search engine^{97–99}. We did not use fixed cutoffs, as the e-value between homologues will vary depending on the protein investigated, hence the need for manual examination of potential homologues and a combination of lines of evidence.

In addition, to identify potential new ESPs, we first used profile-profile searches against EggNOG and manually investigated Asgard orthologous groups which had the best hit to a eukaryotic-specific EggNOG cluster. We also extracted PFAM domains whose taxonomic distribution is exclusive to eukaryotes as per PFAM version 32, and investigated cases where they represented the best domain hit in Asgard archaea sequences identified by HMMscan.

Finally, we manually investigated dozens of proteins known to be involved in key eukaryotic functions based on our knowledge and literature searches. In Figure 2, we are only reporting cases based on the strict cutoff that the diagnostic HMM profile had the best score among all profiles detected for a protein. An exception was made for the ESCRT domain Vps28, Steadiness box, UEV, Vps25, NZF, GLUE and Vps22 domains which are

usually found in combination with other protein domains and thus do not necessarily represent the best scoring domain in a protein even if they represent true homologs.”

R2.5: Here, a comparison with the results of Koonin and Li would be necessary, I think.

This comparison, although justified, is in fact very difficult to do because of the dramatically different approaches taken to identify and define ESP “families”. In the work by Liu and colleagues, because they have used automatic clustering and annotation, they largely tend to split single protein families into many “different” ESPs. For example, in Supplementary Material 9, they report 52 “ESP” (i.e. separate COGs, according to them) as “Longin domain” (which had been reported previously in Spang et al (2015, Nature) and Zaremba et al Niedzwiedska et al (Nature, 2017)) while we report them as a single ESP. The reason for this is that, for many families, it is difficult to reconstruct a resolved phylogeny of each family and pinpoint precisely duplications that predate Asgard archaea and have been inherited in eukaryotes. As such, we think it is more conservative to only report the origin of at least one member of this family from Asgard archaea. In addition, from our experience, each individual genome of Asgard tends to mostly have one (or very few) copies belonging to each ESP superfamily, but clustering algorithms tend to split members of a given family into many different clusters because of the sometimes very uneven evolutionary distance that can exist between homologs of certain groups (such as the long branching Thorarchaeota, or Njordarchaeales). Consequently, these many clusters that are identically functionally annotated often do not represent paralogues but subsets of orthologues. If we collapse the COGs reported by Liu and colleagues that have a similar annotation, they only report about 25 novel ESPs. Out of these, 5 overlap with the ones we had initially reported in Suppl Table 3 but are not discussed in the Liu et al. paper. Moreover, during this revision process, we uncovered several additional ESPs, bringing the number of ESPs reported in our work to 36 (none of which were reported in Liu et al).

R2.6: Concerning your gene-tree/species-tree reconciliation and the reconstruction of the ancestral features of LAECA, I am not sure I understand why you included TACK and Euryarchaeota and not only Asgard? How do your results correlate to the ones by Williams et al on the root of archaea? I do not remember if they had Asgard in that analysis already. This is just a curiosity, and by no means essential to the paper. Line 968 in M&M, please say how you assembled the protein families.

Response: We included homologs from TACK and Euryarchaeota because that allows for a more accurate inference of the gene content of the Last Asgard Common Ancestor (LAsCA) using ancestral genome reconstruction by reconciliation approaches. This is particularly relevant for cases where a protein has a sparse distribution in Asgard archaea but is also present in TACK and/or Euryarchaeota. If one includes TACK and Euryarchaeota in the analyses, one can infer its presence in the LAsCA with more confidence than if the taxon sampling included was only composed of Asgard archaea.

Regarding the comparison with the work by Williams and colleagues: the authors only included a single Asgard representative at the time (one Loki), so they did not have any predictions for the ancestral gene content of Asgard archaea.

The assembly of protein families was done as described in the “Identification of homologous protein families” section of the methods section.

Concerning the inferred higher duplication rates, I am sure many readers will ask if this is not due to incorrect binning and double representation of genes in the reconstructed MAGs. It would be good to mention and exclude this problem in the main text. For example, it could be checked rapidly on the families specifically duplicated in Skadi, if they are 100 identical or contaminations.

Response: We thank the reviewer for bringing up this concern. However, as outlined below, this concern is not justified for the reasons given below:

First of all, the contamination rate estimated by CheckM is not significantly different between the Hodarchaeales (previously proposed “Skadi”) MAGs included in the ALE analyses and the other Heimdallarchaeia (including Njord, previously “Idunn”) (mean=4.3, SD=2.3 versus mean=4.3, SD=2.9); $t(0.025)=5.47$, p-value=0.47).

First, if it was due to contamination or misprediction of genes that would appear artificially duplicated due to incorrect binning, the duplication events would be predicted to be at the tips, i.e. specific to individual MAGs, which we did not take into account to investigate the genome dynamics. In contrast, what we report here are duplications inferred in ancestral nodes and thus common to two or more lineages/MAGs.

Furthermore, if the duplicates were due to contamination, then they would be specific to one of the Hodarchaeales MAGs here again. However, it is true that in the phylogeny, these two copies would not branch together and might lead to inferring an ancient duplication. In this case, because this second copy would be absent from other Hodar, we should observe a high inferred loss rate across Hodar branches, but this is not the case (Figure 4).

In addition, if we compare the estimated contamination level of Heimdallarchaeia MAGs to all other Asgard archaea, it is in fact lower, although not statistically significant (mean=4.3, SD=2.8 versus mean=5.3, SD=7.4; $t(0.025)=2$; p-value=1).

We have briefly mentioned this concern in the main text as follows (and more elaborately in the Supplementary material):

“Importantly, as missing genes and potential contaminations in MAGs will be regarded as recent gene loss and gain events in our ancestral reconstruction analyses, the use of incomplete MAGs with low contamination levels is unlikely to have a major impact on the inferred gene content of the deep archaeal ancestors that were reconstructed in the present study (also see Supplementary Information).”

If this paper is accepted, please be careful to deposit all raw data, not only the MAGs analyzed here. This is very important.

We agree with the reviewer that this is important. All raw data have been deposited in NCBI and will be released upon the manuscript’s publication. Furthermore, all raw data

underlying phylogenomic analyses (raw and processed alignments and corresponding phylogenetic trees) will be deposited on Figshare (<https://figshare.com/account/home#/projects/111912>).

Also, I think you'll need to coordinate with Koonin, Li, and Rinke on how to name these new phyla.

Response: We appreciate the reviewer's comment, and we have indeed made an effort to unite our naming scheme with the ones proposed by and in coordination with other authors. Please see our detailed response to Reviewer #1 (R1.1) on this point.

Referee #3 (Remarks to the Author):

A: Eme, Tamarit, Caceres et al. present a thorough and careful reconstruction of the Asgard archaeal superphylum, including a detailed examination of the branching of Eukaryotes within this group, based on sophisticated methodologies. The authors examine Eukaryotic signature proteins, identifying several new types represented within Asgardian genomes, and reconstruct predictions for ancestral life styles for both LAsCA and LAECA. The manuscript is clearly written and the information presented represents a significant advance in our understanding of the diversity within the Asgard superphylum and the nature of the relationship of the Asgard archaea with the Eukaryotes, from an evolutionary perspective.

Response: We would like to thank the reviewer for evaluating our work.

B: Originality and significance: The work complements recent work examining this radiation and its relationship to Eukaryogenesis, with a more detailed reconstruction of ancestral phenotypes and different ESPs identified. It is a valuable and impressive contribution to a rapidly maturing area of research.

Response: We would like to thank the reviewer for this positive assessment.

C: Data and methodology: The data is handled expertly, and the quality of the data is high. There are several points where the authors draw conclusions that are not fully supported by the data, or which appear selected based on a favoured scenario, which need to be tempered or more strongly justified before the paper would be acceptable for publication.

Response: We appreciate this comment and we hope to have now satisfied this point (see below).

D: Appropriate use of statistics and treatment of uncertainties: Appropriate tests to examine divergences or overlaps (e.g., amino acid frequency data) are used to bolster conclusions. Uncertainty is occasionally under-reported in the main text compared to the supplement, leading to overly strong conclusions.

Response: We appreciate this comment and have made an effort to better convey the uncertainty of the results in the main text, which now reads as:

In summary, resolving the position of eukaryotes relative to Asgard archaea is anything but trivial (see Supplementary Discussion). In our efforts to extract the true phylogenetic signal,

we provide confident support for eukaryotes forming a well-nested clade within the Asgard archaea phylum, consistent with the 2D tree of life scenario. More specifically, we observe that eukaryotes affiliate with the Heimdallarchaeia in analyses in which we systematically reduce phylogenetic artefacts, predominantly converging on a position of eukaryotes as sister to Hodarchaeales, which is also in line with the observed ESP content and genome evolution dynamics (see below).

In addition, we made an effort to explain more clearly why we favor a specific scenario in view of the different lines of evidence in the main text. Finally, our supplementary material contains an extensive discussion about the phylogenomics results, and all trees and alignments are available to the reader.

E: Conclusions: as stated, the general conclusions and new data are robust and exciting. The conclusions around more specific relationships or metabolic reconstructions are in some cases too speculative.

Response: We have revised some of our metabolic inferences in light of the reviewers comments (see below).

F: Suggested improvements: see below for my detailed comments and suggested edits.

G: Referencing is appropriate and complete.

H: Clarity and context: The paper is written clearly and concisely, with substantial supplemental data to bolster the presented work.

My detailed comments follow below:

Major comments:

1) Data is not available. Supplemental table 1 lists the 64 new MAGs, but only 9 have MAG-related accessions, and none of the Biosample accessions for the other MAGs that I checked were accessible on NCBI. These should have been available at the time of submission.

Response: We made the MAGs available to reviewers as a Figshare archive, the link to which is in the supplementary material. We apologize if this was not obvious enough. The data has been deposited ahead of submission to NCBI and will be released upon acceptance for publication.

2) Originality and novelty: There have been several recent papers/pre-prints expanding the Asgard radiation and naming new clades. Recent publications expanding the Asgard lineage and naming phyla require a reassessment of the proposed names in this work – are any of the Gefion-, Freya-, Vidar-, and Idunnarchaeota the same lineages as described in Liu et al., Nature 2021 (<https://www.nature.com/articles/s41586-021-03494-3>) and if so, please adjust nomenclature accordingly.

I appreciate that these genomes were not available at the time of submission, but it would be irresponsible not to assess overlaps in novel clades prior to the formal namings presented in the supplemental materials being accepted.

Response: We have now included these data and revised our naming and proposed the unification of the Asgard taxonomic scheme. Please see our response to reviewer 1 (R1.1).

3) How are phyla being determined? From phylogenies, 11 lineages of Asgard does not appear commensurate with phylum-level lineages in other archaeal radiations, by sequence distance. Idunn is clearly a new group, but the others are less clear aside from inconsistent associations with other lineages – the rationale for phylum determination is not presented.

Response: We agree with this and made an effort to homogenize ranking with what has been defined for other archaea (also see our response to reviewers above). In particular, we have leaned on the efforts by Rinke et al (2021, Nat Microbiol 6: 946) and their application to Asgard archaeal taxonomy in Sun et al. (2021, ISME Comm 1: 30) and recent versions of the Genome Taxonomy Database. In particular, Asgard archaea are now considered a single phylum here (see Extended Data Figure 1) comprised of a number of classes corresponding to most of the groups previously considered as phyla.

4) The instability of the Asgard lineages with respect to each other suggests resolving power is limited and the deeper divergence to the Euks also unstable – it is difficult to understand why a single answer (sister to Skadiarchaea) was selected as correct given the robust trees under alternate hypotheses are equally well supported/poorly supported, and the phyloBayes did not converge – that alone suggests there is not sufficient information to resolve the specific branch point. This is a major point of the paper, and I think the conclusion (the title of the paper) is overstepping the data.

Response: We agree with the reviewer that across our analyses the positions of various Asgard clades and eukaryotes are not always stable. However, our systematic phylogenomic analyses aimed to reduce phylogenetic artefacts, combined with the observed ESP distribution and genome dynamic patterns, indicate that Hodarchaeales are likely to represent the closest relatives to eukaryotes. However, we recognize that the phylogenetic signal is convoluted and that our work will likely not be the workd in the debate. We have now carefully reworded our conclusion and title to reflect that the one consistent and clear result is that eukaryotes emerged well nested with Asgard archaea, with their closest relatives being Heimdallarchaeia. Yet, we state that the most predominant support is obtained for eukaryotes as being sister to Hodarchaeales.

Similarly: Supplemental materials Page 27, line 585 – did the Phylobayes for NM57 datasets converge? If not, what were the topologies observed along the different chains?

Response: No, it turned out computationally infeasible to have the chains converge. However, the topologies from the different chains run for variations of the NM57 dataset were all very consistent. They all show eukaryotes forming a monophyletic clade with Heimdallarchaeia (including Njord). The detail for the support given by each chain is shown in Supp Table 2 (see column “(H,E) for the support for Heimdallarchaeia and Eukaryotes).

The treatment of the datasets and phylogenetic signals seems unequal – lack of convergence is listed as a reason to doubt the RP56 data, but not the NM57.

Response: The reviewer raises a fair point. The lack of convergence for the Bayesian analyses of the RP56 dataset was however not in itself the reason to question the result,

but instead the fact that the position of Njord (previously, Idunn) changes dramatically between chains: some chains show Njord to be either sister to Korarchaeota or, in stark contrast, forming a clade with Heimdallarchaeia (e.g., chains from RP56-A64-nD, Fig. S20, or chains from RP56-A64-nDE, Suppl Table 2). In contrast, and although the chains did not converge either for the NM57 dataset, they all showed Njord to be monophyletic with (or to cluster within) Heimdallarchaeia. With the expanded taxon sampling (A175), the runs for the RP56 datasets are much more convergent, and Njord now consistently branch within Heimdallarchaeia across all chains. However, the backbone of Asgard archaea and the position of eukaryotes are still unresolved across these chains, while they are resolved, strongly supported and congruent across the chains ran on the NM57 dataset (see Supp Table 2).

Euks are unresolved across RP56 treatments, but equally unresolved in NM57 cases, where one treatment is selected as the correct inference. Stronger justification for selecting one option as correct is needed throughout. The justification of Korarchaeota/Idunnarchaeota attraction based on amino acid composition is sufficient for those decisions, but does not speak to the placement of Euks in trees experiencing that artifact.

Response: It is true that the amino acid bias observed in the RP56 dataset seems to affect Njord and Korarchaeota, but not the eukaryotic sequences. However, because eukaryotes are related to the Njord/Heimdall clade, and because Njordarchaeales are the longest-branching lineage within Heimdallarchaeia, there is a strong side effect of Njord being misplaced as sister to Korarchaeota, by attracting eukaryotes outside Asgard archaea, as sister to Njord, or to Njord+Korarchaeota. The overall distortion that the compositional bias imposes on the tree thus makes the RP56 suboptimal to investigate the placement of eukaryotes. Furthermore, the justification for combining data treatments (recoding and fast evolving site removal) for the placement of eukaryotes is because analyses of individual treatments obtained insufficiently supported or inconsistent results, indicating that the true phylogenetic signal to for placing eukaryotes was compromised by phylogenetic artefacts (e.g. due to compositional bias or mutational saturation). By combining treatments, we obtained more consistent and better supported phylogenetic signal, indicating that eukaryotes are a sister group to Hodarchaeales.

5) the CIII complex in Skadiarchaea – activity in conjunction with a Rieske center is highly speculative, and based on one protein undertaking a role it is not normally associated with (Narl passing electrons to Rieske iron-sulfur protein) and a poorly resolved prediction – “potential Rieske iron sulfur protein”, which is also suggested to play a role in electron transfer from Narl to cupredoxin. Cupredoxin itself has not been seen as an electron shuttle in archaea, adding another layer of conjecture to this reconstruction. This, of all hypothesized metabolic and energetic functions, is the least convincing, and is featured in main text figure 4. I feel this is too speculative to be given such weight in the main document.

We have rephrased the main text of the manuscript to yield a more cautious view of our metabolic reconstructions. We have removed mention of a bona fide complex III in the Hodarchaeales (Skadi) ancestor. Instead, we present a hypothesis for a nitrate reductase-like complex that may utilize cupredoxin-type electron carriers. This is based on previous observations that showed some nitrous oxide reductases that can accept electrons from cupredoxin electron carriers (doi/10.1073/pnas.0711316105).

6) Throughout, there are different datasets in use, and the descriptions of these datasets are not clearly presented. I recognize this is a collaborative work, likely with different aspects occurring in parallel, but which dataset is which, and which of the novel MAGs were included in each is not well telegraphed in the text.

Examples:

Page 6 line 64 – why only 17 additional Asgard MAGs? There are more available, and for deep tree reconstructions, shorter branches are advantageous. What constituted “phylogenetically diverse”? If these were selected specifically for 16S rRNA genes associated with MAGs, that should be stated clearly. Your later dataset (page 7, line 90) contained 64 Asgard MAGs before the addition of the new 63 MAGs, so this disconnect between dataset completion is confusing.

Response: We have now clarified the dataset used for each analysis by giving them specific names, and whose qualities are summarized in Supplementary Discussion and Supplementary Table 2.

We have also clarified what we meant by phylogenetically diverse. The methods now read as: “An updated Asgard archaeal genomic sequence dataset was constructed by including all 230 Asgard archaeal MAGs and genomes available at the NCBI database as of May 12, 2021, as well as 63 novel MAGs described in the present work. All 56 trimmed RP alignments were concatenated into an RP56-A293 supermatrix (465 taxa including 293 Asgard archaea, 7112 amino acid positions), which was used to infer a preliminary phylogeny with FastTree v2 (Supplementary Figure 16). Given the high computational demands of the subsequent analyses, we then used this phylogeny to select a subsample of Asgard archaea representatives. For this, we first removed the most incomplete MAGs encoding fewer than 19 ribosomal proteins (i.e., 1/3 of the markers) in the matrix. We also used the preliminary phylogeny to sub-select among closely related taxa: among taxa that were separated by branch lengths of <0.1 , we only kept one representative. This led to a selection of 331 genomes, including 175 Asgard archaea, 41 DPANN, 43 Euryarchaeota, and 72 TACK representatives (RP56-A175 dataset). Out of these 175 Asgard archaea, 41 correspond to MAGs newly reported here. Once this taxon set was gathered, we identified homologs of the NM57 gene set as described above, thus generating supermatrix NM57-A175. All datasets and their composition are summarized in Supplementary Table 2.”

Note that the selection is the selection made based on the RP-A293 phylogeny is visible in Supplementary Figure 16.

Page 7 line 90 – updated by including additional MAGs, or some other form of updating? If addition, please clarify if the numbers of MAGs stated for the different groups are the current or original set. If current, why only 64 MAGs for Asgard, and were these pre-existing publicly available MAGs, or the newly described MAGs from this study, or a mix?

Response: Indeed, we kept the marker selection but changed the taxon sampling by adding (and removing) taxa. We have only kept the previously identified homologs for the 9 Asgard archaea representatives included in Zaremba-Niedzwiedzka et al. 2017. For all other archaeal representatives, we did a *de novo* search for orthologues as described in the Methods section.

Now, with the inclusion of the more recently published Asgard MAGs, these numbers have changed significantly, as reflected in the main text. For Asgard archaea, we retrieved homologs from a total of 293 representatives (as described previously): 63 of these represent the new MAGs described in this paper.

However, in order to reduce the size of the dataset, we have removed a few closely related taxa (as described just above), to end up with 175 Asgard archaea in the dataset used for most phylogenomic analyses. Out of these, 49 are from our new MAGs.

Page 39, line 736 – If all archaeal genomes were downloaded and one representative per species-level cluster was used, how are the three archaeal lineages acting as an outgroup? They are not outside the Archaea. Did you mean all Asgard archaeal genomes?

Please make explicit which genomes were added to this dataset from your new MAGs, as there is some confusion from the main text as to dataset sizes and overlaps.

Response: The goal of this analysis was to investigate the diversity of Asgard archaeal MAGs by means of a phylogenetic reconstruction of ribosomal proteins encoded in the same contig. In this context, sequences from representatives of the TACK superphylum (in particular, those that are named Bathyarchaeia, Nitrososphaeria and Thermoproteia in GTDB) were used as outgroup, as opposed to the Asgard archaeal sequences, which were our ingroup.

Page 40, lines 749-754 – additional detail is needed here to clarify what the dataset underlying this tree is – the addition of the NCBI Asgard to the previous dataset? Is this an entirely new dataset, with a different reference set or outgroup?

Response: We thank the reviewer for bringing this up. We realized that our original presentation and wording were confusing. We have now streamlined this by only including a tree containing our Asgard MAGs, Asgard MAGs from NCBI, and outgroup sequences (i.e. archaea from the DPANN, TACK and Euryarchaeota), and clarified the Methods (lines 904-925)

Page 42, line 786 – which dataset is this – the original tree, the NCBI-Asgard-included tree, or another? How many of those Asgard MAGs were from this study? I suggest developing names for your different datasets, and making explicit what is contained in each one.

Response: We recognize the confusion here and have followed the reviewer’s suggestion to give more explicit names to the different datasets and what they contain. This section now reads “All-versus-all similarity searches of all predicted proteins from the A64 taxon selection (64 Asgard, 76 TACK, 43 Euryarchaeota and 41 DPANN archaea) were performed using diamond blastp (--more-sensitive --evaluate 0.0001 --max-target-seqs 0 --outfmt 6).”

7) Page 48, line 918 – what is the justification for: “These protein pairs were joined by concatenating the amino acid sequence of the proteins containing beta-propeller repeats (N-terminus) and alpha-solenoid repeats (C-terminus)” for proteins that are up to 10 genes apart from each other on the genome? Is there evidence of such distant interactions for other proteins within these (or other) genomes? Are the genes operonic with each other? This is a stretch compared to the other protein families examined, especially given “individual β -propeller and α -solenoid folds are commonly found among prokaryotes” (supp material page 21, line 453). Are there similar pairs in other, non-Asgard, genomes that are co-located and receive similar support under a concatenated protein structure analysis?

Response: Our reasoning was that, in archaeal genomes, co-regulated genes tend to share gene neighborhoods, while the exact synteny of such regions is not necessarily maintained. However, we agree with the reviewer that the evidence for gene neighbourhood co-location resulting in protein interaction is weak, and have decided to remove these results from our manuscript.

Supplemental figure 25 – the gene neighbourhoods do not dispel my concerns around positing protein fusions/complexing for these genes, which are largely non-operonic.

Response: Agreed. See above.

Minor comments

Main text:

Page 13 line 228 – give the median for % genome for small GTPases in place of, or at least, alongside the maximum.

Response: Here, we were referring to results from previously published work. We have not calculated this proportion for the new set of MAGs.

Page 13 line 238 – missing the word “of”: of the gamma subunit.

Response: Thank you, we have corrected this.

Page 15 – is the increase in proteome size for the Skadia solely due to gene duplications? What role is LGT afforded in your models?

Response: No, the genome increase in Hodarchaeales (former Skadiarchaeia) is the result of bot gene duplication, transfer and origination events. For example, in the branch towards the last common ancestor of Hodarchaeales (node 223 in Supplementary Figure 12) 195 duplications, 427 transfers and 236 originations were inferred, and also 701 losses (see Supplementary Table 8 for inferred values).

Page 16 line 307 – your data supports this prediction, it does not confirm it.

Response: We have corrected this.

Page 19 line 365 – sentence starting “Of these, Asgard archaeal proteins...” is difficult to follow. Please revise.

Response: We have rephrased this.

Page 20 ,line 381 – consider if “maintained” is the correct word here?

Response: In principle this is correct, but we have rephrased as “which seem to generally contain”.

Figure 1 line 514 – using “at least 5 proteins” out of 15 targeted is below the recommended sequence completion for a concatenated gene alignment – ideally taxa have 50% presence within alignment sites, otherwise long-branch effects are artefactually observed.

Response: We respectfully disagree with this. It has been shown that large amounts of missing data do not affect phylogenetic reconstruction (Philippe, H., et al. "Pitfalls in supermatrix phylogenomics." European Journal of Taxonomy 283 (2017)). Moreover the RP15 tree is largely congruent with our RP56 tree, suggesting that artefacts, if they exist, are not due to missing data (although these proteins likely carry similar compositional issues than those of the RP56 concatenation). Finally, 72% of the taxa included in the RP15 dataset were represented by 8 or more ribosomal proteins. All in all, we argue that the RP15 phylogeny presented here is sufficient for the initial investigation of the Asgard archaeal diversity represented by our MAGs.

Figure 2 – what was the rationale for selected Asgard MAG inclusion here? Only a small fraction of the expanded dataset is used, and while I appreciate the rigor of the phylogenetic analyses, and the presentation of alternate, supported relationships, the taxon selection needs better justification.

Response: We hope to have clarified our rationale in our previous responses. Furthermore, it is worth noting that our selection includes 60% (175/293) of all published Asgard MAGs as of May 2021.

Figure 4 – “The Wood-Ljungdahl pathway (WLP) appeared only to be present in the LAsCA and was lost in the more recent ancestors examined here.” – consider rephrasing, as you examine all Asgard lineages so “here” could mean the manuscript, or only Heimdall and Idunn in this figure. As written it implies WLP is missing from all extant Asgard lineages, and that makes reconstruction of the ancestor overly speculative.

Response: Thank you for pointing out this lack of clarity. We have rephrased as follows: “The Wood-Ljungdahl pathway (WLP) appeared only to be present in the LAsCA and was lost in the more recent ancestors (of Heimdallarchaeia and Hodarchaeales) indicated here”.

Methods

Page 41 lines 780-783 – how were annotations from different servers integrated?

Response: We have used these annotations as preliminary evidence for the function of gene families that were manually investigated and discussed throughout the manuscript. But these were then manually curated. There was no hard rule as to how to integrate these annotations. We took into account consensus across tools, statistical support for said annotations, etc. and carried out a number of manual checks. The methods now state:

“In-depth analysis of potential ESPs involved a combination of automatic screens and manual curation. We first manually searched for homologs of previously described ESPs9,10,42 by using a variety of sequence similarity approaches such as BLAST, HMMer tools, profile-profile searches using HHblits, combined with phylogenetic inferences, and, in some cases, the Phyre2 structure homology search engine103,110,111. We did not use fixed cutoffs, as the e-value between homologs will vary depending on the protein investigated, hence the need for manual examination of potential homologs and a combination of lines of evidence.

In addition, to identify potential new ESPs, we first used our profile-profile searches against EggNOG and manually investigated Asgard orthologous groups which had a best hit to a eukaryotic-specific EggNOG cluster. We also extracted PFAM domains whose taxonomic distribution is exclusive to eukaryotes as per PFAM v32, and investigated cases where they represented the best domain hit in Asgard archaea sequences identified by HMMscan. Finally, we manually investigated dozens of proteins known to be involved in key eukaryotic functions based on our knowledge and literature searches. In Figure 2, we are only reporting cases based on the strict cutoff that the diagnostic HMM profile had the best score among all profiles detected for a protein. An exception was made for the ESCRT domain Vps28, Steadiness box, UEV, Vps25, NZF, GLUE and Vps22 domains which are usually found in combination with other protein domains and thus do not necessarily represent the best scoring domain in a protein even if they represent true homologs.”

Page 43, line 816 – sentence/phrase starting “we kept only the ones..” is grammatically incorrect, I think some words are out of order.

Response: Thank you, we have rephrased that section.

Fast site removal – trialed 10-90% removal, which was used for the final tree? I’m not sure that was stated explicitly.

Response: That is correct, we have now made this more explicit in each figure legend. Additionally, the support values for each of the removal steps are reported in Supplementary Table 2.

Page 48 – Ancestral reconstruction – as before, please make explicit which genomes were included, what role the newly described MAGs play in this analysis, and how genomes were selected for inclusion.

Response: We have now clarified this in the Methods, all genomes used for this analysis are listed in Supplementary Table 2. “For the ancestral reconstruction analyses, only a subset of 181 taxa were included (64 Asgard, 74 TACK and 43 Euryarchaeota, see Supplementary Table 2 for details).”

Page 49 line 939 - is the threshold of 0.3 a proportion or a different statistic? Inferring presence ancestrally with 30% presence within a family seems low. Is there precedence for this threshold? Also, the sentence says 0.3 “for an event to be considered” – how were decisions ultimately made?

What were the empirical requirements? On line 943 it also states “considered if its copy number was above 0.3” – considered, or accepted? Considered implies a later decision process that is not described.

Response: The threshold of 0.3 was used as it was the recommended threshold by the authors of the ALE software (Gergely Szöllösi), and this has been empirically tested and applied in Dharamshi et al 2023 *Nature Microbiology*, 8 40-54. We agree that the phrasing of “considered” is confusing. This should be “accepted” (not “considered”). We have rephrased this in the manuscript, and now use “scored” to make it clearer:

“Metabolic reconstruction of the Asgard ancestors was based on the inference, annotation and copy number of genes in ancestral nodes. The presence of a given gene was scored if its copy number in the ancestral nodes was above 0.3. A protein family was scored as “maybe present” if the inferred copy number was between 0.1 and 0.3. The protein annotation of each of the clusters containing the ancestral nodes was manually verified for each of the enzymatic steps involved in the pathways detailed in Supplementary Information.”

Supplemental materials

Page 22, line 480 – font change, this section also lacks references where appropriate.

Response: We have corrected this.

Page 24, line 517 – similarly, references or data (figure mention) lacking for Yip section

Response: We have corrected this.

Page 25 line 554 – what makes a BAR domain a “clear BAR domain” – threshold of similarity? Scores? Please clarify.

Response: In the methods section we describe how ESPs were identified (lines 973-992). Furthermore, we have omitted “clear” from the description of the BAR domain.

Page 33, line 736 – sentence “As expected, several ‘patchy’ ESPs currently only found in few Asgard archaeal genomes, such as tubulin (only present in Odinararchaeota genomes), are predicted to be absent in LAECA-proxy nodes, such as Sec23/Sec24, TRAPP and ubiquitin-like proteins.” Has two sets of examples, interspersing the main sentence – is confusing, suggest rephrasing. Are Sec23/Sec24 etc. the LAECA-proxy nodes? I think they are not, but rather examples of patchy ESPs – not as written currently.

Response: We agree this phrasing is confusing. During the revision, we have removed this section from the supplementary discussion.

Page 34 line 757 – “Other homologous protein families” – how many? Given you provide 4 examples, and all 4 are then inferred to be present – how many were not considered likely present in LAECA?

Response: It is not straightforward to extract such numbers for the ancestral inference analyses. Please note that this phrase has been removed this section from the manuscript.

Page 35 line 784 – “nearly most” is not informative, provide a proportion or precise number

Response: We have corrected this.

Page 44, line 1000 – “A recent study...” – reference is missing.

This reference has been added (De Anda et al, *Nature communications*, 12, 2404)

Page 49 line 1142 – “could not identified detect” – remove identified.

Response: We have corrected this.

Supplemental figure 9 is much less well-formatted than the preceding trees.

Response: We have corrected this, and in addition have provided separate high resolution versions for all Supplementary Figures in a separate PDF file (see zip file ‘Supplementary_figures_2021-04-06336R1.zip’).

Supplemental figure 11 is completely unreadable – the resolution prohibits any meaningful examination of the trees presented.

Response: We apologize for this, we have now provided the figures as separate PDF files and not simply embedded in the Word document (see zip file ‘Supplementary_figures_2021-04-06336R1.zip’).

Supplemental figure 17 – what are the scales of the dots (content/originations)? Missing from legend.

Response: We have now provided a modified version of this figure with information on the size of the dots in the legend (see current Suppl Fig 12).

Supplemental figure 23 has space for the bin names/scaffold names using location codes – the symbols are very small and difficult to distinguish even at maximum zoom, plus they do not add clarity to the figure.

Response: We have now modified this figure by removing these symbols.

Supplemental figure 32 is not described clearly enough to interpret. I have tried, and I genuinely do not know what numbers correspond to what comparisons/statistics, or what the trend line (?) on scatter plots are, among other confusions. Needs clearer axes, full figure legend.

Response: We apologize for the confusion brought by this figure. Upon reflection, we decided to remove it and only report the relevant coefficients and p-values for the Spearman correlations discussed in the text (Supplementary Table 6).

Reviewer Reports on the First Revision:

Referees' comments:

Referee #2 (Remarks to the Author):

I congratulate the authors for their titanic effort on this revised version of the manuscript, which contains a compelling case for the most updated and solid phylogeny of Asgard without and with eukaryotes, an updated and consensus taxonomy, and a rich supplementary material with in depth description of the results. I have no further comments than being happy to see it finally out.

Referee #3 (Remarks to the Author):

The authors have completed a significant, but informative revision of their previous manuscript, including updating their datasets to include newly described MAGs for lineages of the Asgard, and adapting their proposed nomenclature as needed. The revisions are thorough, and have taken my concerns around tree selection and data clarity into account. I have no further revision suggestions, and congratulate the authors on an interesting and important paper.